# Single-loop Algorithms for Stochastic Non-Convex Optimization with Weakly-Convex Constraints

**Ming Yang**                                                        *myang@tamu.edu*
*Department of Computer Science & Engineering*
*Texas A&M University*

**Gang Li**                                                          *gang-li@tamu.edu*
*Department of Computer Science & Engineering*
*Texas A&M University*

**Quanqi Hu**                                                        *quanqi-hu@tamu.edu*
*Department of Computer Science & Engineering*
*Texas A&M University*

**Qihang Lin**                                                       *qihang-lin@uiowa.edu*
*Tippie College of Business*
*The University of Iowa*

**Tianbao Yang**                                                     *tianbao-yang@tamu.edu*
*Department of Computer Science & Engineering*
*Texas A&M University*

**Reviewed on OpenReview:** *https://openreview.net/forum?id=aCgOR2KvAI*

## Abstract

Constrained optimization with multiple functional inequality constraints has significant applications in machine learning. This paper examines a crucial subset of such problems where both the objective and constraint functions are weakly convex. Existing methods often face limitations, including slow convergence rates or reliance on double-loop algorithmic designs. To overcome these challenges, we introduce a novel single-loop penalty-based stochastic algorithm. Following the classical exact penalty method, our approach employs a **hinge-based penalty**, which permits the use of a constant penalty parameter, enabling us to achieve a **state-of-the-art complexity** for finding an approximate Karush-Kuhn-Tucker (KKT) solution. We further extend our algorithm to address finite-sum coupled compositional objectives, which are prevalent in artificial intelligence applications, establishing improved complexity over existing approaches. Finally, we validate our method through experiments on fair learning with receiver operating characteristic (ROC) fairness constraints and continual learning with non-forgetting constraints.

## 1 Introduction

This paper focuses on solving the following non-linear inequality constrained optimization problem:

$$\min_{\mathbf{x}} F(\mathbf{x}), \quad \text{s.t.} \quad h_k(\mathbf{x}) \leq 0, \quad k = 1, \dots, m, \tag{1}$$

where $F$ and $h_k$, $k = 1, \dots, m$ are stochastic, weakly convex and potentially non-smooth. This general formulation captures a wide range of practical optimization problems where non-smoothness arises in the objective and constraint functions. Inequality constrained optimization arise in many applications in ML/AI, including continual learning (Li et al., 2024a), fairness-aware learning (Vogel et al., 2021), multi-class Neyman-Pearson classification (mNPC) problem (Ma et al., 2020), robust learning (Robey et al., 2021), policy optimization problem for reinforcement learning (Schulman et al., 2015), and distributed data center scheduling problem (Yu et al., 2017).

Table 1: Comparison with existing works. We consider that both objective function and constraint function oracles are stochastic. In the Constraint column, $h_k$ means multiple constraints, $k$ from 1 to $m$. In Convexity column, NC means Non-convex, WC means Weakly Convex, C means Convex. The monotonicity column means non-decreasing. The last column Complexity means the complexity for finding an (nearly) $\epsilon$-KKT point of equation 1.

| Reference | Loop | Objective | Constraints | Smoothness | Convexity | Monotonicity | Complexity |
|---|---|---|---|---|---|---|---|
| (Alacaoglu & Wright, 2024) | Single Loop | $F(\cdot)$ | $h(\cdot) = 0$ | $F, h$ | NC $(F, h)$ | none | $\tilde{\mathcal{O}}(\epsilon^{-5})$ |
| (Li et al., 2024b) | Double Loop | $F(\cdot) + g(\cdot)$ | $h(\cdot) = 0$ | $F, h$ | C $(g)$, NC $(F, h)$ | none | $\tilde{\mathcal{O}}(\epsilon^{-5})$ |
| (Ma et al., 2020) | Double Loop | $F(\cdot)$ | $h_k(\cdot) \leq 0$ | none | WC $(F, h_k)$ | none | $\mathcal{O}(\epsilon^{-6})$ |
| (Boob et al., 2023) | Double Loop | $F(\cdot)$ | $h_k(\cdot) \leq 0$ | none | WC $(F, h_k)$ | none | $\mathcal{O}(\epsilon^{-6})$ |
| (Huang & Lin, 2023) | Single Loop | $F(\cdot)$ | $h(\cdot) \leq 0$ | none | WC $(F)$, C $(h)$ | none | $\mathcal{O}(\epsilon^{-8})$ |
| (Li et al., 2024a) | Single Loop | $\sum_i f_i(g_i(\cdot))$ | $h_k(\cdot) \leq 0$ | $f_i, g_i, h_k$ | NC $(f_i, g_i, h_k)$ | none | $\mathcal{O}(\epsilon^{-7})$ |
| (Liu & Xu, 2025) | Single Loop | $F(\cdot)$ | $h_k(\cdot) \leq 0$ | none | WC $(F, h_k)$ | none | $\mathcal{O}(\epsilon^{-6})$ |
| Ours | Single Loop | $F(\cdot)$ | $h_k(\cdot) \leq 0$ | none | WC$(F, h_k)$ | none | $\mathcal{O}(\epsilon^{-6})$ |
| Ours | Single Loop | $\sum_i f_i(g_i(\cdot))$ | $h_k(\cdot) \leq 0$ | none | WC $(f_i, g_i, h_k)$ | $f_i$ | $\mathcal{O}(\epsilon^{-6})$ |
| Ours | Single Loop | $\sum_i f_i(g_i(\cdot))$ | $h_k(\cdot) \leq 0$ | $f_i, g_i$ | WC $(h_k)$ | none | $\mathcal{O}(\epsilon^{-6})$ |

In optimization literature, many techniques have been studied for solving convex constrained optimization problems, including primal-dual methods (Huang et al., 2025; Boob et al., 2023; Pu et al., 2024; Nedić & Ozdaglar, 2009), augmented Lagrangian methods (Birgin & Martnez, 2014), penalty methods (Boyd & Vandenberghe, 2004), and level-set methods (Lin et al., 2018). However, the presence of non-convexity in the problem often complicates algorithmic design and convergence analysis, while non-smoothness and stochasticity add additional layers of difficulty.

Several recent studies have explored stochastic non-convex constrained optimization under various assumptions. Ma et al. (2020) and Boob et al. (2023) initiated the study of stochastic optimization with weakly convex and non-smooth objective and constraint functions. Inspired by proximal-point methods for weakly-convex unconstrained optimization (Davis & Grimmer, 2019), they have proposed a double-loop algorithm, where a sequence of convex quadratically regularized subproblems is solved approximately. They have derived the state-of-the-art complexity in the order of $O(1/\epsilon^6)$ for finding a nearly $\epsilon$-KKT solution under a uniform Slater's condition and a strong feasibility condition, respectively.

Recently, Alacaoglu & Wright (2024); Li et al. (2024a); Huang & Lin (2023) have proposed single-loop algorithms for stochastic non-convex constrained optimization. However, these results either are restricted to smooth problems with equality constraints (Alacaoglu & Wright, 2024) or suffer from an iteration complexity higher than $O(1/\epsilon^6)$ (Li et al., 2024a; Huang & Lin, 2023). This raises an intriguing question: *"Can we design a single-loop method for solving non-smooth weakly-convex stochastic constrained optimization problems to match the state-of-the-art complexity of $O(1/\epsilon^6)$ for finding a nearly $\epsilon$-KKT solution?"*

We answer the question in the affirmative. We propose a novel algorithm based on the well-studied exact penalty method (Zangwill, 1967; Evans et al., 1973). Different from Alacaoglu & Wright (2024); Li et al. (2024a) who used a quadratic or squared hinge penalty, we construct a penalty of the constraints using the **non-smooth hinge function**, i.e., by adding $\frac{1}{m} \sum_{k=1}^{m} \beta [h_k(\mathbf{x})]_+$ into the objective, where $\beta$ is a penalty parameter. Despite being non-smooth, the benefits of this penalty function include (i) it preserves the weak convexity of in the penalty; (ii) it allows using a constant $\beta$ as opposed to a large $\beta$ required by the squared hinge penalty (Li et al., 2024a); (iii) it introduces the structure of finite-sum coupled compositional optimization (FCCO), enabling the use of existing FCCO techniques in the algorithmic design. Consequently, this approach establishes state-of-the-art complexity for achieving a nearly $\epsilon$-KKT solution, under a regularity assumption similar to those in Li et al. (2024a); Alacaoglu & Wright (2024). We summarize our contributions as follows.

- We propose a non-smooth, hinge-based exact penalty method for tackling non-convex constrained optimization with weakly convex objective and constraint functions. We derive a theorem to guarantee that a nearly $\epsilon$-stationary solution of the penalized objective is a nearly $\epsilon$-KKT solution of the original problem under a regularity condition of the constraints.

- We develop algorithms based on FCCO to tackle a non-smooth weakly convex objective whose unbiased stochastic gradient is available and a structured non-smooth weakly-convex objective that

is of the FCCO form. We derive a state-of-the-art complexity of $\mathcal{O}(\epsilon^{-6})$ for finding a nearly $\epsilon$-KKT solution.

- We conduct experiments on two applications in machine learning: learning with ROC fairness constraints and continual learning with non-forgetting constraints. The effectiveness of our algorithms are demonstrated in comparison with an existing method based on squared hinge penalty function and a double-loop method.

## 2 Related Work

Despite their long history of study, recent literature on optimization with non-convex constraints has introduced new techniques and theories for the penalty method (Lin et al., 2022), Lagrangian methods (Sahin et al., 2019; Sun & Sun, 2024; Li et al., 2024b; 2021), the sequential quadratic programming methods (Berahas et al., 2021; 2025; Curtis et al., 2024; Na et al., 2023) and trust region method (Fang et al., 2024). These works mainly focused on the problems with smooth objective and constraint functions. On the contrary, based on Moreau envelopes, Ma et al. (2020); Boob et al. (2023); Huang & Lin (2023); Jia & Grimmer (2022) have developed algorithms and theories for problems with non-smooth objective and constraint functions. However, these methods cannot be applied when the objective function has a compositional structure. The method by Li et al. (2024a) can be applied when the objective function has a coupled compositional structure, but it requires the objective and constraint functions to be smooth. Compared to their work, we relax the smoothness assumption, but assume the monotonicity of the outer function in the compositional structure.

Alacaoglu & Wright (2024) studies a single-loop quadratic penalty method for stochastic smooth nonconvex equality constrained optimization under a generalized full-rank assumption on the Jacobian of the constraints. Under a similar assumption, Li et al. (2024b) proposed a double-loop inexact augmented Lagrangian method using the momentum-based variance-reduced proximal stochastic gradient algorithm as a subroutine. Both Alacaoglu & Wright (2024) and Li et al. (2024b) consider **only smooth equality constraints and hence are not applicable to non-smooth constraint functions**. Ma et al. (2020) and Boob et al. (2023) studied weakly convex non-smooth optimization with inequality constraints. They proposed double-loop methods, where a sequence of proximal-point subproblems is solved approximately by a stochastic subgradient method (Yu et al., 2017) or a constraint extrapolation method (Boob et al., 2023). Under a uniform Slater's condition and a strong feasibility assumption, respectively, Ma et al. (2020) and Boob et al. (2023) establish a complexity of $\mathcal{O}(1/\epsilon^6)$. Under the uniform Slater's condition, Huang & Lin (2023) proposed a single-loop switching subgradient method for non-smooth weakly convex constrained optimization problems, but the complexity of their method is $\mathcal{O}(1/\epsilon^8)$ when the constraints are stochastic.

Motivated by applications from continual learning with non-forgetting constraints, Li et al. (2024a) introduced a single-loop squared hinge penalty method for a FCCO (Wang & Yang, 2022) problem, where the objective function takes the form of $\sum_i f_i(g_i(\cdot))$ and the constraints are non-convex and smooth. They establish a complexity of $\mathcal{O}(1/\epsilon^7)$ under the generalized full-rank assumption on the Jacobian of constraints. Our results can be easily extended to a FCCO problem with nonconvex but unnecessarily smooth inequality constraints. We show that our method improves the complexity to $\mathcal{O}(1/\epsilon^6)$ in two cases: (1) $f_i$, $g_i$ and $h_k$ are non-smooth weakly convex and $f_i$ is monotone non-decreasing, and (2) $f_i$ and $g_i$ are smooth nonconvex but $h_k$ is non-smooth weakly convex. Note that the second case includes the special case where $f_i$, $g_i$ and $h_k$ are smooth non-convex. A comparison with prior results in presented in Table 1.

We also note that some works have considered transferring inequality constraints into equality constraints by adding a slack variable for each constraint (Fukuda & Fukushima, 2017; Ding & Wright, 2023; Li et al., 2021). However, these approaches either require stronger second-order conditions to ensure that a KKT solution to the equality-constrained problem also applies to the original inequality-constrained problem (Fukuda & Fukushima, 2017; Ding & Wright, 2023), or they rely on the boundedness of constraint functions to guarantee this transfer (Li et al., 2021) (cf. Section 4.3 for more discussions). Even when such a transformation is feasible, algorithms for non-smooth stochastic constraint functions remain underdeveloped, underscoring the uniqueness and significance of our contributions.

Finally, we would like to acknowledge a concurrent work by Liu & Xu (2025). Their study also explores a penalty method for problems with weakly convex objective and constraint function. The key differences

between our work and theirs are: (i) although they consider a vector-valued function consisting of multiple components, they treat multiple constraint as a single one, i.e., they need to compute mini-batch estimators of all constraint functions at each iteration in their algorithm. In contrast, we do sampling on the constraint functions, and only require estimating $O(1)$ constraint functions at each iteration. (ii) they do not directly use the hinge penalty function, but instead adopt a smoothed version of it; and (iii) they employ the SPI-DER/SARAH technique to track the constraint function value, which requires periodically using large batch sizes. In contrast, we track multiple constraint function values by employing the MSVR technique (Jiang et al., 2022) using only a constant batch size.

## 3 Notations and Preliminaries

Let $\|\cdot\|$ be the $\ell_2$-norm. For $\psi : \mathbb{R}^d \to \mathbb{R} \cup \{+\infty\}$, the subdifferential of $\psi$ at $\mathbf{x}$ is $\partial\psi(\mathbf{x}) = \{\zeta \in \mathbb{R}^d \mid \psi(\mathbf{x}') \geq \psi(\mathbf{x}) + \langle\zeta, \mathbf{x}'-\mathbf{x}\rangle + o(\|\mathbf{x}'-\mathbf{x}\|), \mathbf{x}' \to \mathbf{x}\}$, and $\zeta \in \partial\psi(\mathbf{x})$ denotes a subgradient of $\psi$ at $\mathbf{x}$. Let $[\cdot]_+ := \max\{0, \cdot\}$ denote the hinge function. We say $\psi$ is $\mu$-strongly convex ($\mu \geq 0$) if $\psi(\mathbf{x}) \geq \psi(\mathbf{x}') + \langle\zeta, \mathbf{x}-\mathbf{x}'\rangle + \frac{\mu}{2}\|\mathbf{x}-\mathbf{x}'\|^2$ for any $(\mathbf{x}, \mathbf{x}')$ and any $\zeta \in \partial\psi(\mathbf{x}')$. We say $\psi$ is $\rho$-weakly convex ($\rho > 0$) if $\psi(\mathbf{x}) \geq \psi(\mathbf{x}') + \langle\zeta, \mathbf{x}-\mathbf{x}'\rangle - \frac{\rho}{2}\|\mathbf{x}-\mathbf{x}'\|^2$ for any $(\mathbf{x}, \mathbf{x}')$ and any $\zeta \in \partial\psi(\mathbf{x}')$. We say $\psi$ is $L$-smooth ($L \geq 0$) if $|\psi(\mathbf{x}) - \psi(\mathbf{x}') + \langle\nabla\psi(\mathbf{x}'), \mathbf{x}-\mathbf{x}'\rangle| \leq \frac{L}{2}\|\mathbf{x}-\mathbf{x}'\|^2$ for any $(\mathbf{x}, \mathbf{x}')$. For simplicity, we abuse $\partial\psi(x)$ to denote one subgradient from the corresponding subgradient set when no confusion could be caused.

Throughout the paper, we make the following assumptions on (1)

**Assumption 3.1.** (a) $F$ is $\rho_0$-weakly convex and $L_F$-Lipschitz continuous; (b) $h_k(\cdot)$ is $\rho_1$-weakly convex and $L_h$-Lipschitz continuous for $k = 1, \ldots, m$.

For a non-convex optimization problem, finding a globally optimal solution is intractable. Instead, a Karush-Kuhn-Tucker (KKT) solution is of interest, which is an extension of a stationary solution of an unconstrained non-convex optimization problem. A solution $\mathbf{x}$ is a KKT solution to (1) if there exist $\boldsymbol{\lambda} = (\lambda_1, \ldots, \lambda_m)^\top \in \mathbb{R}_+^m$ such that $0 \in \partial F(\mathbf{x}) + \sum_{k=1}^m \lambda_k \partial h(\mathbf{x})$, $h_k(\mathbf{x}) \leq 0, \forall k$ and $\lambda_k h_k(\mathbf{x}) = 0 \ \forall k$. Of interest in non-asymptotic analysis is finding an $\epsilon$-KKT solution given below.

**Definition 3.2.** A solution $\mathbf{x}$ is an $\epsilon$-KKT solution to (1) if there exist $\boldsymbol{\lambda} = (\lambda_1, \ldots, \lambda_m)^\top \in \mathbb{R}_+^m$ such that $\text{dist}(0, \partial F(\mathbf{x}) + \sum_{k=1}^m \lambda_k \partial h(\mathbf{x})) \leq \epsilon$, $h_k(\mathbf{x}) \leq \epsilon, \forall k$, and $\lambda_k h_k(\mathbf{x}) \leq \epsilon$ if $h_k(\mathbf{x}) \geq 0$ or $\lambda_k = 0$ if $h_k(\mathbf{x}) < 0$ for $k = 1, \ldots, m$.

However, since the objective and the constraint functions are non-smooth, finding an $\epsilon$-KKT solution is not tractable, even the constraint functions are absent. Let us consider a simple example $\min_{\mathbf{x}} |\mathbf{x}|$. The only stationary point is the optimal solution $\mathbf{x}_* = 0$, and any $\mathbf{x} \neq 0$ is not an $\epsilon$-stationary solution ($\epsilon < 1$) no matter how close $\mathbf{x}$ to 0. In other words, unless the iterate generated by the algorithm can exactly land on $\mathbf{x} = 0$, it will not produce an $\epsilon$-stationary point. To address this issue, an effective approach for solving non-smooth optimization is to approximate the original problem by a smoothed one using different smoothing techniques, including Nesterov's smoothing(Nesterov, 2005), randomized smoothing (Kornowski & Shamir, 2022), and Moreau envelope (Davis & Grimmer, 2019). Based on Moreau envelopes, Ma et al. (2020); Boob et al. (2023); Huang & Lin (2023); Jia & Grimmer (2022) have developed algorithms and theories for problems with non-smooth objective and constraint functions, which derives a nearly $\epsilon$-KKT solution. We follow Moreau envelopes technique on non-smooth weakly convex optimization and consider finding a nearly $\epsilon$-KKT solution defined below.

**Definition 3.3.** A solution $\mathbf{x}$ is a nearly $\epsilon$-KKT solution to (1) if there exist $\bar{\mathbf{x}}$ and $\boldsymbol{\lambda} = (\lambda_1, \ldots, \lambda_m)^\top \in \mathbb{R}_+^m$ such that (i) $\|\mathbf{x} - \bar{\mathbf{x}}\| \leq O(\epsilon)$, $\text{dist}(0, \partial F(\bar{\mathbf{x}}) + \sum_{k=1}^m \lambda_k \partial h(\bar{\mathbf{x}})) \leq \epsilon$, (ii) $h_k(\bar{\mathbf{x}}) \leq \epsilon, \forall k$, and (iii) $\lambda_k h_k(\bar{\mathbf{x}}) \leq \epsilon$ if $h_k(\bar{\mathbf{x}}) \geq 0$ or $\lambda_k = 0$ if $h_k(\bar{\mathbf{x}}) < 0$ for $k = 1, \ldots, m$.

In order to develop our analysis, we introduce the Moreau envelope of a $\rho$-weakly convex function $\phi$: $\phi_\theta(\mathbf{x}) := \min_{\mathbf{y}} \left\{\phi(\mathbf{y}) + \frac{1}{2\theta}\|\mathbf{y} - \mathbf{x}\|^2\right\}$, where $\theta < \rho^{-1}$. The Moreau envelope is an implicit smoothing of the original problem (cf. Lemma B.1 in Appendix). Hence, if we find $\mathbf{x}$ such that $\|\nabla\phi_\theta(\mathbf{x})\| \leq \epsilon$, then we can say that $\mathbf{x}$ is close to a point $\bar{\mathbf{x}}$ that is $\epsilon$-stationary (Davis & Grimmer, 2019). We call $\mathbf{x}$ a nearly $\epsilon$-stationary solution of $\min_{\mathbf{x}} \phi(\mathbf{x})$.

**Remark:** Our current formulation focuses on the non-smooth case, there is also a simple extension in the smooth setting. Suppose the objective and constraint functions are smooth, using the Lipschitz contiuity of the gradients, and that of constraint functions, it is easy to prove that a nearly $\epsilon$-stationary point is also

an $O(\epsilon)$-stationary point. This is because $\|\mathbf{x} - \bar{\mathbf{x}}\|_2 \leq \mathcal{O}(\epsilon)$ and leveraging the Lipschitz conunity condition, every condition in the definition of nearly $\epsilon$-KKT solution can be converted to that on $\mathbf{x}$.

# 4  A Hinge Exact Penalty Method and Theory

The key idea of the hinge-based exact penalty method is to solve the following unconstrained minimization (Zangwill, 1967; Evans et al., 1973), where $\beta > 0$ is a penalty parameter:

$$\min_{\mathbf{x}} \Phi(\mathbf{x}) := F(\mathbf{x}) + \frac{\beta}{m} \sum_{k=1}^{m} [h_k(\mathbf{x})]_+, \tag{2}$$

For clarity of notation, we define $h_k^+(\mathbf{x}) = [h_k(\mathbf{x})]_+$. Let $\partial[h_k(\mathbf{x})]_+ \in [0, 1]$ be the subdifferential of $[z]_+$ at $z = h_k(\mathbf{x})$, and let $\partial h_k^+(\mathbf{x})$ be the subgradient in terms of $\mathbf{x}$.

**Lemma 4.1.** *Under Assumption 3.1, $\Phi(\mathbf{x})$ in (2) is $C$-weakly convex and $L$-Lipschitz, where $C = \rho_0 + \beta\rho_1$ and $L = L_F + \beta L_h$.*

Our hinge exact penalty method is to employ a stochastic algorithm for finding an $\epsilon$-stationary solution to $\Phi_\theta(\mathbf{x})$ for some $\theta < 1/C$. We establish a theorem below to guarantee that such $\mathbf{x}$ is a nearly $\epsilon$-KKT solution to (1).

**Theorem 4.2.** *Assume that there exists $\delta > 0$ such that*

$$dist\left(0, \frac{1}{m}\sum_{k=1}^{m} \partial h_k^+(\mathbf{x}')\right) \geq \delta, \forall \mathbf{x}' \in \mathcal{V} \tag{3}$$

*where $\mathcal{V} = \{\mathbf{x} : \max_k h_k(\mathbf{x}) > 0\}$. If $\mathbf{x}$ is an $\epsilon$-stationary solution to $\Phi_\theta(\mathbf{x})$ for some $\theta < 1/C$ and a constant $\beta > (\epsilon + L_F)/\delta$, then $\mathbf{x}$ is a nearly $\epsilon$-KKT solution to the original problem (1).*

*Proof.* Let $\bar{\mathbf{x}} = \text{prox}_{\theta\Phi}(\mathbf{x})$. By optimality of $\bar{\mathbf{x}}$, we have

$$0 \in \partial F(\bar{\mathbf{x}}) + \frac{\beta}{m}\sum_{k=1}^{m} \partial h_k^+(\bar{\mathbf{x}}) + (\bar{\mathbf{x}} - \mathbf{x})/\theta.$$

Since $\|\nabla\Phi_\theta(\mathbf{x})\| \leq \epsilon$, then we have $\|\bar{\mathbf{x}} - \mathbf{x}\| = \theta\|\nabla\Phi_\theta(\mathbf{x})\| \leq \theta\epsilon$. Then we have

$$\text{dist}\left(0, \partial F(\bar{\mathbf{x}}) + \frac{\beta}{m}\sum_{k=1}^{m} \partial h_k^+(\bar{\mathbf{x}})\right) \leq \|(\bar{\mathbf{x}} - \mathbf{x})/\theta\| \leq \epsilon.$$

Since $\partial h_k^+(\bar{\mathbf{x}}) = \xi_k \partial h_k(\bar{\mathbf{x}})$ (see Bauschke et al. (2017, Corollary 16.72)), where

$$\xi_k = \begin{cases} 1 & \text{if } h_k(\bar{\mathbf{x}}) > 0, \\ [0,1] & \text{if } h_k(\bar{\mathbf{x}}) = 0, \quad \subseteq \partial[h_k(\bar{\mathbf{x}})]_+, \\ 0 & \text{if } h_k(\bar{\mathbf{x}}) < 0, \end{cases}$$

there exists $\lambda_k \in \frac{\beta\xi_k}{m} \geq 0, \forall k$ such that $\text{dist}\left(0, \partial F(\bar{\mathbf{x}}) + \sum_{k=1}^{m} \lambda_k \partial h_k(\bar{\mathbf{x}})\right) \leq \epsilon$. Thus, we prove condition (i) in Definition 3.3. Next, let us prove condition (ii). We argue that $\max_k h_k(\bar{\mathbf{x}}) \leq 0$. Suppose this does not hold, i.e., $\max_k h_k(\bar{\mathbf{x}}) > 0$, we will derive a contradiction. Since $\exists \mathbf{v} \in \partial F(\bar{\mathbf{x}})$ such that

$$\epsilon \geq \text{dist}\left(0, \mathbf{v} + \frac{\beta}{m}\sum_{k=1}^{m} \partial h_k^+(\bar{\mathbf{x}})\right) \geq \text{dist}\left(0, \frac{\beta}{m}\sum_{k=1}^{m} \partial h_k^+(\bar{\mathbf{x}})\right) - \|\mathbf{v}\| \geq \beta\delta - L_F, \tag{4}$$

which is a contradiction to the assumption that $\beta > (\epsilon + L_F)/\delta$. Thus, $\max_k h_k(\bar{\mathbf{x}}) \leq 0$. This proves condition (ii). The last condition (iii) holds because: $\lambda_k = \frac{\beta\xi_k}{m}$, which is zero if $h_k(\bar{\mathbf{x}}) < 0$; and $\lambda_k h_k(\bar{\mathbf{x}}) \leq 0$. $\qquad \square$

Assumption (3) is not specific to our method but is shared by all penalty-based approaches, including recent works such as Alacaoglu & Wright (2024), Li et al. (2024b), and Li et al. (2024a). This assumption reflects a fundamental challenge in non-convex constrained optimization: without a condition such as (3), the problem becomes ill-posed and is generally unsolvable using first-order methods. There is potential to slightly relax assumption (3). In our current formulation, we assume (3) holds over the entire set $\mathcal{V}$. However, it may suffice to require (3) to hold only on a subset of $\mathcal{V}$, defined as $\mathcal{V}_c := \{x : c \geq \max_k h_k(x) > 0\}$ for some constant $c > 0$. This is a strictly weaker assumption.

Under this relaxation, the algorithm can be initialized within the feasible region and proceed with step sizes chosen carefully to ensure that all intermediate iterates remain within $\mathcal{V}_c$. Note that even if some

iterates become slightly infeasible, as long as the step size is sufficiently small, the gradients of the constraint functions will guide the iterates back toward feasibility. This way, we can still conduct the same convergence analysis just select feasible initial point $\mathbf{x}_0$ and proceed with step sizes chosen carefully. This strategy has been successfully adopted in prior work (e.g., (Huang & Lin, 2023)).

### 4.1 Sufficient Conditions for Eq. (3)

Next, we provide some sufficient conditions for (3).

First, we consider a simple setting where there is only one constraint $h(\mathbf{x}) \leq 0$. The lemma below states two conditions sufficient for proving the condition (3). The proof of Lemma 4.3 and 4.4 are presented in Appendix B.

**Lemma 4.3.** *If there exists $\mu > 0, c > 0$ such that (i) $h(\mathbf{x}) - \min_{\mathbf{y}} h(\mathbf{y}) \leq \frac{1}{2\mu} dist(0, \partial h(x))^2$ for any $\mathbf{x}$ satisfying $h(\mathbf{x}) > 0$; and (ii) $\min_{\mathbf{y}} h(\mathbf{y}) \leq -c$, then condition (3) holds for $\delta = \sqrt{2c\mu}$.*

**Remark:** Note that the condition (i) in the above lemma is the PL condition of $h$ but only for points violating the constraint. We can easily come up a function $h$ to satisfy the two conditions in the lemma. Let us consider $h(x) = |x^2 - 1| - 1$. Then $\nabla h(0) = 0$, which means $x = 0$ is a stationary point but not global optimum of $h$. The global optimal are $x \in \{1, -1\}$ and $\min_y h(y) = -1$, and hence condition (ii) holds. Any $x$ such that $h(x) > 0$ would satisfy $x > \sqrt{2}$ or $x < -\sqrt{2}$. For these $x$, we have $\nabla h(x) = 2x$ for $x > \sqrt{2}$ or $\nabla h(x) = -2x$ for $x < -\sqrt{2}$. Then for $|x| > \sqrt{2}$, $h(x) - \min_y h(y) = x^2 - 1 \leq \frac{|\nabla h(x)|^2}{2\mu}$ holds for $\mu = 2$. Hence condition (i) holds.

Next, we present a sufficient condition for (3) for multiple constraints. The following lemma shows that the full rank assumption of the Jacobian matrix of $H(\mathbf{x}) = (h_1(\mathbf{x}), \ldots h_m(\mathbf{x}))^\top$ at $\mathbf{x}$ that violate constraints is a sufficient condition.

**Lemma 4.4.** *If $\partial H(\mathbf{x})$ is full rank for $\max_k h_k(\mathbf{x}) > 0$, i.e., $dist(0, \lambda_{\min}(\partial H(\mathbf{x}))) \geq \sigma > 0, \forall \mathbf{x}$ such that $\max_k h_k(\mathbf{x}) > 0$, where $\partial H(\mathbf{x})$ denotes the set of sub-Jacobian matrices and $\lambda_{\min}(\cdot)$ denotes the set of their minimum singular values, then the condition (3) holds for $\delta = \sigma/m$.*

We refer to the above condition as Full-Rank-at-Violating-Points condition (FRVP). We note that the FRVP condition for differentiable constraint functions has been used as a sufficient condition in Li et al. (2024a) for ensuring

$$\|\nabla H(\mathbf{x})^\top [H(\mathbf{x})]_+\| \geq \delta \|[H(\mathbf{x})]_+\|, \forall \mathbf{x}, \tag{5}$$

which is central to proving the convergence of their squared-hinge penalty method for solving inequality constrained smooth optimization problems: $\min_{\mathbf{x}} F(\mathbf{x})$, s.t. $h_k(\mathbf{x}) \leq 0, k = 1, \ldots, m$.

A stronger condition for equality constrained optimization with $h_k(\mathbf{x}) = 0, k = 1, \ldots, m$ was imposed in Alacaoglu & Wright (2024), where they assumed that $\|\nabla H(\mathbf{x})^\top H(\mathbf{x})\| \geq \delta \|H(\mathbf{x})\|, \forall \mathbf{x}$. To ensure this, their method requires the full-rank Jacobian assumption to hold at all points, whereas our approach only requires it at constraint-violating points. In our experiments, we verified that the condition in Lemma 4.4 holds on the trajectory of the algorithm. Detailed results are provided in Table 2 in Appendix A.

### 4.2 Issue with squared-hinge penalty

Let us discuss why using the squared-hinge penalty function as in Li et al. (2024a). For simplicity of exposition and comparison with prior works, we consider differentiable constraint functions here. Using a squared-hinge penalty function gives the following objective:

$$\Phi(\mathbf{x}) = F(\mathbf{x}) + \frac{\beta}{m} \sum_{k=1}^{m} [h_k(\mathbf{x})]_+^2. \tag{6}$$

Note that this objective is also weakly convex with a weak-convexity parameter given by $\rho_0 + \beta\rho_1$. However, the key issue of using this approach is that $\beta$ needs to be very large to find an (nearly) $\epsilon$-KKT solution. Using the same argument as in the proof of Theorem 4.2 for showing $\max_k h_k(\bar{\mathbf{x}}) \leq 0$, the inequality (4)

becomes:

$$\epsilon \geq \left\| \mathbf{v} + \frac{\beta}{m} \sum_{k=1}^{m} [h_k(\bar{\mathbf{x}})]_+ \nabla h_k(\bar{\mathbf{x}}) \right\| \geq \left\| \frac{\beta}{m} \sum_{k=1}^{m} [h_k(\bar{\mathbf{x}})]_+ \nabla h_k(\bar{\mathbf{x}}) \right\| - \|\mathbf{v}\| \geq \frac{\beta\delta}{m} \|[H(\bar{\mathbf{x}})]_+\| - L_F,$$

where the last step follows the assumption in (5). Since $[H(\bar{\mathbf{x}})]_+$ could be very small, in order to derive a contradiction we need to set $\beta$ to be very large. In Li et al. (2024a), it was set to be the order of $O(1/\epsilon)$. As a result, $\Phi(\cdot)$ will have large weak-convexity and Lipschitz parameters of the order $O(1/\epsilon)$. This will lead to a complexity of $O(1/\epsilon^{12})$ if we follow existing studies(Hu et al., 2024) to solve the problem in (6) without the smoothness of $F$ and $h_k$, which will be discussed in Section 5.

### 4.3  Issue of solving an equivalent equality constrained optimization

One may consider the following equivalent equality constrained problem:

$$\min_{\mathbf{x}, s_1, \ldots, s_m} F(\mathbf{x}), \quad \text{s.t.} \quad h_k(\mathbf{x}) + s_k^2 = 0, k = 1, \ldots, m \tag{7}$$

where $s_1, \ldots, s_m$ are introduced dummy variables. A solution $\mathbf{x}^*$ is a KKT solution to above if there exists $\boldsymbol{\lambda} = (\lambda_1, \ldots, \lambda_m)^\top \in \mathbb{R}^m$ such that $0 \in \partial F(\mathbf{x}^*) + \sum_{k=1}^{m} \lambda_k \partial h_k(\mathbf{x}^*)$, $\lambda_k s_k = 0, \forall k$, and $h_k(\mathbf{x}^*) + s_k^2 = 0$ for $\forall k = 1, \ldots, m$. However, this does not provide a guarantee that $\mathbf{x}_*$ is a KKT condition to the original constrained optimization problem as $\boldsymbol{\lambda}$ is not necessarily non-negative. For instance, consider $\min x$ s.t. $-1 \leq x \leq 1$. After adding slack variables, we have $\min x$ s.t. $x + s_1^2 = 1, -x + s_2^2 = 1$. However, $(x, s_1, s_2) = (1, 0, \sqrt{2})$ becomes a stationary point with multipliers $\lambda = (-1, 0)$, but it is not a stationary point in the original problem.

While several studies (Fukuda & Fukushima, 2017; Ding & Wright, 2023) have employed stronger second-order conditions to ensure that a KKT solution of the equality-constrained problem (7) can be converted into one for the original inequality-constrained problem, these conditions generally do not hold for the problems we consider. Moreover, when these second-order conditions fail to be satisfied, the approach of solving the equality-constrained problem (7) becomes ineffective. Our solutions will not suffer from this issue.

Specifically, Proposition 3.6. in Fukuda & Fukushima (2017) shows that if a stationary solution to the equality-constrained problem (formulated using squared slack variables) satisfies the **second-order sufficient condition (SOSC)**, then it is also a stationary solution to the original inequality-constrained problem (after simply removing the slack variables). Theorem 3.3. in reference (Ding & Wright, 2023) proves a similar result but under the weaker **second-order necessary condition (SONC)**. To the best of our knowledge, no other conditions have been established in the literature.

Both SOSC and SONC conditions above are quite strong, as they require all first-order stationary points of the equality-constrained problem **to be also second-order stationary** points at the same time. Unfortunately, neither SOSC nor SONC is implied by the assumptions made in our paper. This is true even if we further assume our problem is second-order differentiable. Since our focus is on first-order methods, our assumption is related to first-order conditions only. In fact, in general, our problems may not even have gradients (non-smooth), let alone the Hessian matrix in SOSC and SONC. While the results in Fukuda & Fukushima (2017); Ding & Wright (2023) may be useful in practice as potential certificates of stationarity, their applicability is limited in practice. For example, one could first find a stationary point for the equality-constrained problem, and then verify whether SOSC or SONC holds **at the obtained solution**. If lucky enough and they hold, one can conclude that the solution is stationary for the original inequality-constrained problem. However, if unlucky and these second-order conditions do not hold, we do not know if the solution is stationary for the original problem. In contrast, our method directly finds a stationary point for the inequality-constrained problem without requiring such verification, hence there is no need to implement the verification procedure.

Another way is to use non-negative slack variables as following:

$$\min_{\mathbf{x}, s_1, \ldots, s_m \geq 0} F(\mathbf{x}), \quad \text{s.t.} \quad h_k(\mathbf{x}) + s_k = 0, k = 1, \ldots, m \tag{8}$$

where $s_1, \ldots, s_m$ are introduced dummy variables. Li et al. (2021) presents some discussion how to convert an $\epsilon$-KKT solution to the above equality constrained into an $\epsilon$-KKT solution to the original inequality constrained optimization. However, such guarantee requires that boundness of the constraint functions.

---

**Algorithm 1** *Algorithm for solving equation 2 under Setting I*

---
1: **Initialization:** choose $\mathbf{x}_0, \beta, \gamma_2$ and $\eta$.
2: **for** $t = 0$ to $T - 1$ **do**
3:     Sample $\mathcal{B}_c^t \subset \{1, \ldots, m\}$
4:     **for** each $k \in \mathcal{B}_c^t$ **do**
5:         Sample a mini-batch $\mathcal{B}_{2,k}^t$
6:         Update $u_{2,k}^{t+1} = (1 - \gamma_2)u_{2,k}^t + \gamma_2 h_k(\mathbf{x}_t; \mathcal{B}_{2,k}^t) + \gamma_2'(h_k(\mathbf{x}_t; \mathcal{B}_{2,k}^t) - h_k(\mathbf{x}_{t-1}; \mathcal{B}_{2,k}^t))$
7:     **end for**
8:     Let $u_{2,k}^{t+1} = u_{2,k}^t, k \notin \mathcal{B}_c^t$
9:     Compute $G_2^t = \frac{\beta}{|\mathcal{B}_c^t|} \sum_{k \in \mathcal{B}_c^t} \partial h_k(\mathbf{x}_t; \mathcal{B}_{2,k}^t)[u_{2,k}^t]_+', G_1^t = \frac{1}{|\mathcal{B}^t|} \sum_{\zeta \in \mathcal{B}^t} \partial f(\mathbf{x}_t; \zeta)$.
10:     Update $\mathbf{x}_{t+1} = \mathbf{x}_t - \eta(G_1^t + G_2^t)$
11: **end for**

---

## 5  Stochastic Algorithms

In this section, we consider stochastic algorithms for solving (2) for $F(\mathbf{x}) = \mathbb{E}_\zeta[f(\mathbf{x}, \zeta)]$, which has been considered in Ma et al. (2020); Huang & Lin (2023); Boob et al. (2023). Due to limit of space, we present extension and analysis of compositional objective $F(\mathbf{x}) = \frac{1}{n} \sum_{i=1}^n f(\mathbb{E}_\zeta[g_i(\mathbf{x}, \zeta)])$ in the Appendix D. Without loss of generality, we assume the constrained functions are only accessed through stochastic samples. Below, we let $h(\mathbf{x}, \mathcal{B}) = \frac{1}{|\mathcal{B}|} \sum_{\xi \in \mathcal{B}} h(\mathbf{x}, \xi)$ with a mini-batch of samples $\mathcal{B}$. We make the following assumptions regarding $h_k$, and $f$.

**Assumption 5.1.** For any $\mathbf{x}$, assume there exists $h_k(\mathbf{x}, \xi)$, and $\partial h_k(\mathbf{x}, \xi)$, where $\xi$ is random variable, such that (a) $\mathbb{E}[h_k(\mathbf{x}, \xi)] = h_k(\mathbf{x})$ and $\mathbb{E}[\partial h_k(\mathbf{x}, \xi)] \in \partial h_k(\mathbf{x})$; (b) $\mathbb{E}_\xi[|h_k(\mathbf{x}, \xi) - h_k(\mathbf{y}, \xi)|^2] \leq L_h^2 \|\mathbf{x} - \mathbf{y}\|^2$; (c) $\mathbb{E}_\xi[|h_k(\mathbf{x}, \xi) - h_k(\mathbf{x})|^2] \leq \sigma_h^2$.

**Assumption 5.2.** For any $\mathbf{x}$, assume there exists $\partial f(\mathbf{x}, \zeta)$, where $\zeta$ is random variable such that $\mathbb{E}[\partial f(\mathbf{x}, \zeta)] \in \partial F(\mathbf{x})$ and $\mathbb{E}_\zeta[\|\partial F(\mathbf{x}) - \partial f(\mathbf{x}, \zeta)\|^2] \leq \sigma_f^2$.

The key to our algorithm design is to notice that the hinge-based exact penalty function $\frac{\beta}{m} \sum_{k=1}^m [h_k(\mathbf{x})]_+$ is a special case of non-smooth weakly-convex finite-sum coupled compositional objective as considered in Hu et al. (2024), where the inner functions are $h_k(\mathbf{x})$ and the outer function is the hinge function that is monotone and convex. Hence, we can directly utilize their technique for computing a stochastic gradient estimator of the penalty function.

The subgradient of the penalty function is given by $\frac{1}{m} \sum_{k=1}^m \beta \partial [h_k(\mathbf{x})]_+ \partial h_k(\mathbf{x})$. However, computing $h_k(\mathbf{x})$ and $\partial h_k(\mathbf{x})$ is prohibited due to its stochastic nature or depending on many samples. To this end, we need an estimator for $h_k(\mathbf{x})$ and $\partial h_k(\mathbf{x})$. $\partial h_k(\mathbf{x})$ can be simply estimated by its mini-batch estimator. The challenge is to estimate $h_k(\mathbf{x})$ using mini-batch samples. The naive approach that estimates $h_k(\mathbf{x})$ by its mini-batch estimator does not yield a convergence guarantee for compositional optimization. To address this issue, we use the variance reduction technique MSVR for estimating constraint functions $h_k(\mathbf{x}), \forall k$ (Jiang et al., 2022), which maintains and updates an estimator $u_{2,k}$ for each constraint function $h_k$. We present the key steps in Algorithm 1. In particular, at the $t$-th iteration, we construct a random mini-batch $\mathcal{B}_c^t \subset \{1, \ldots, m\}$, then we update the estimators of $h_k(\mathbf{x}_t), k \in \mathcal{B}_c^t$ by Step 6 where $\gamma_2 \in (0, 1), \gamma_2' = \frac{m - |\mathcal{B}_c|}{|\mathcal{B}_c|(1 - \gamma_2)} + 1 - \gamma_2$, and $\mathcal{B}_{2,k}^t$ is a mini-batch of samples for estimating $h_k(\mathbf{x}_t)$. Then the gradient of the penalty function and the objective function at $\mathbf{x}_t$ can be estimated by $G_2^t, G_1^t$ in Step 10, where $[u_{2,k}^t]_+'$ is the derivative of the hinge function at $u_{2,k}^t$. Then, we can update $\mathbf{x}_{t+1}$ by using SGD with $G_1^t + G_2^t$ as the stochastic gradient estimator.

Since the estimator $u_{2,k}$ produced by MSVR is used to approximate the true constraint value during optimization, it is useful to clarify how it behaves over iterations. Our analysis of Lemma B.3 shows that as long as $\sqrt{\gamma_2} \to 0$, $\eta/\sqrt{\gamma_2} \to 0$, and $t \to \infty$, the error $|u_{2,k}^t - h_k(\mathbf{x}_t)|$ converges to zero. This guarantees that the solution converges to a KKT point asymptotically. For the non-asymptotic case with $\epsilon > 0$, although $u_{2,k}^t$ may fluctuate around zero near the feasibility boundary, such oscillations can be effectively controlled by choosing sufficiently small $\eta$ and $\gamma_2$. This ensures that the hinge-based constraint $h_k(\mathbf{x}_t) \leq \epsilon$ is reliably enforced. In our experiments, we did not observe any instability in constraint satisfaction.

The convergence of the algorithm for finding an $\epsilon$-stationary solution $\Phi_\theta(x)$ is given below. The proof of the theorem is presented in Appendix C.

**Theorem 5.3.** *Suppose Assumptions 3.1 5.1 and 5.2 hold. Let $\gamma_2 = \mathcal{O}(\frac{|\mathcal{B}_{2,k}|\epsilon^4}{\beta^4})$, $\eta = \mathcal{O}(\frac{|\mathcal{B}_c||\mathcal{B}_{2,k}|^{1/2}\epsilon^4}{\beta^5 m})$, after $T = \mathcal{O}(\frac{\beta^6 m}{|\mathcal{B}_c||\mathcal{B}_{2,k}|^{1/2}\epsilon^6})$ iterations, the Algorithm 1 satisfies $\mathbb{E}[\|\nabla\Phi_\theta(\mathbf{x}_{\hat{t}})\|] \le \epsilon$ from some $\theta = O(1/(\rho_0 + \rho_1\beta))$, where $\hat{t}$ selected uniformly at random from $\{1, \ldots, T\}$.*

**Remark**: While we state the convergence results for a given $\epsilon$, it is also possible to state the convergence for a given $T$. In this case, we can let $\eta, \gamma_2$ depend on $T$, which can yield the same complexity result. In practice, these hyper-parameters are always tuned to achieve the best performance. For example, in the first experiment we choose the initial learning rate from $\{10^{-3}, 10^{-4}\}$ and apply a standard decay schedule; in the second experiment we use the same magnitude choices as in Li et al. (2024a). These settings are fully reported in the experimental section, and our results show that they work well in practice.

Combining the results from Theorem 5.3 and 4.2, we demonstrate Algorithm 1 can find a nearly $\epsilon$-KKT solution to the original problem (1) in the following Corollary. The proof is provided at the end of Appendix C.

**Corollary 5.4.** *Suppose Assumptions 3.1, 5.1, 5.2 and equation 3 hold. Let $\gamma_2 = \mathcal{O}(\frac{|\mathcal{B}_{2,k}|\epsilon^4}{\beta^4})$, $\eta = \mathcal{O}(\frac{|\mathcal{B}_c||\mathcal{B}_{2,k}|^{1/2}\epsilon^4}{\beta^5 m})$ and a constant $\beta > \frac{\epsilon + L_F}{\delta}$, after $T = \mathcal{O}(\frac{\beta^6 m}{|\mathcal{B}_c||\mathcal{B}_{2,k}|^{1/2}\epsilon^6})$ iterations, the Algorithm 1 can find a nearly $\epsilon$-KKT solution $\mathbf{x}_{\hat{t}}$ where condition (i) in Definition 3.3 holds in expectation, conditions (ii) and (iii) in Definition 3.3 hold with probability $1 - \mathcal{O}(\epsilon)$ for the original problem (1). And if $|h_k(\bar{\mathbf{x}}_{\hat{t}})| < +\infty$, the conditions (ii) and (iii) hold in expectation. $\hat{t}$ is selected uniformly at random from $\{1, \ldots, T\}$.*

**Remark:** Corollary 5.4 shows that Algorithm 1 can find a nearly $\epsilon$-KKT solution to the original problem (1) with $T = O(\frac{1}{\epsilon^6})$ iterations, which matches the state-of-the-art complexity of double-loop algorithms (Ma et al., 2020; Boob et al., 2023). It is better than the complexity of $O(1/\epsilon^8)$ of the single-loop algorithm proposed in Huang & Lin (2023). It is noticed that Liu & Xu (2025) also achieve the same complexity of $O(\frac{1}{\epsilon^6})$. While both works achieve comparable complexity guarantees, our contribution lies in a different algorithmic design and problem setting: We directly operate on the hinge penalty instead of a smoothed surrogate. Specifically, our MSVR estimator is designed for estimating many sequences while only accessing $O(1)$ sequence for update. This makes it greatly suitable for us to handle many constraint functions. In contrast, the SPIDER estimator requires accessing all sequences at the same iteration, which is not efficient for handling many constraint functions.

# 6 Applications in Machine Learning

## 6.1 AUC Maximization with ROC Fairness Constraints

We consider learning a model with ROC fairness constraints (Vogel et al., 2021). Suppose the data are divided into two demographic groups $\mathcal{D}_p = \{(\mathbf{a}_i^p, b_i^p)\}_{i=1}^{n_p}$ and $\mathcal{D}_u = \{(\mathbf{a}_i^u, b_i^u)\}_{i=1}^{n_u}$, where $\mathbf{a}$ denotes the input data and $b \in \{1, -1\}$ denotes the class label. A ROC fairness is to ensure the ROC curves for classification of the two groups are the same. Since the ROC curve is constructed with all possible thresholds, we follow Vogel et al. (2021) by using a set of thresholds $\Gamma = \{\tau_1, \cdots, \tau_m\}$ to define the ROC fairness. For each threshold $\tau$, we impose a constraint that the false positive rate (FPR) and true positive rate (TPR) of the two groups are close, formulated as the following:

$$h_\tau^+(\mathbf{w}) := \left| \frac{1}{n_p^+} \sum_{i=1}^{n_p} \mathbb{I}\{b_i^p = 1\}\sigma(s_\mathbf{w}(\mathbf{a}_i^p) - \tau) - \frac{1}{n_u^+} \sum_{i=1}^{n_u} \mathbb{I}\{b_i^u = 1\}\sigma(s_\mathbf{w}(\mathbf{a}_i^u) - \tau) \right| - \kappa \le 0$$

$$h_\tau^-(\mathbf{w}) := \left| \frac{1}{n_p^-} \sum_{i=1}^{n_p} \mathbb{I}\{b_i^p = -1\}\sigma(s_\mathbf{w}(\mathbf{a}_i^p) - \tau) - \frac{1}{n_u^-} \sum_{i=1}^{n_u} \mathbb{I}\{b_i^u = -1\}\sigma(s_\mathbf{w}(\mathbf{a}_i^u) - \tau) \right| - \kappa \le 0,$$

where $s_\mathbf{w}(\cdot)$ denotes a parameterized model, $\sigma(z)$ is the sigmoid function, and $\kappa > 0$ is a tolerance parameter. For the objective function, we use a pairwise AUC loss:

$$F(\mathbf{w}) = \frac{-1}{|\mathcal{D}_+||\mathcal{D}_-|} \sum_{\mathbf{x}_i \in \mathcal{D}_+} \sum_{\mathbf{x}_j \in \mathcal{D}_-} \sigma(s(\mathbf{w}, \mathbf{x}_i) - s(\mathbf{w}, \mathbf{x}_j)),$$

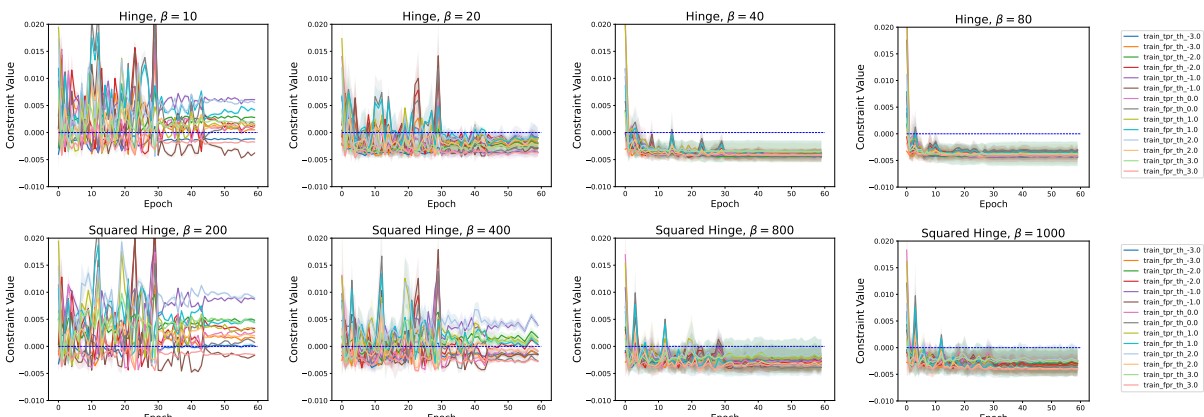

Figure 1: Training curves of 15 constraint functions of different methods for fair learning on the Adult dataset. Top: hinge-based penalty method with different $\beta$; Bottom: squared-hinge-based penalty method with different $\beta$. Each curve (e.g., legend `train_tpr_th_-3.0`) denotes a constraint function $h_{\tau^+}(\mathbf{w})$ (resp. with $\tau = 3$); similarly, `train_fpr_th_-3.0` represents the constraint function $h_{\tau^-}(\mathbf{w})$ with $\tau = 3$.

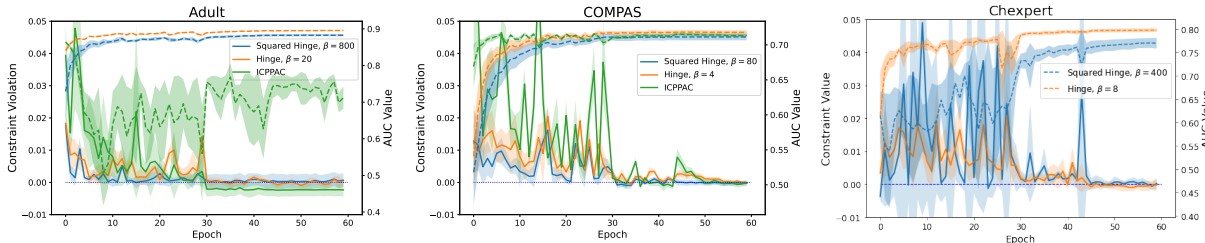

Figure 2: Training curves of objective and constraint violation of different methods under a parameter setting when they satisfy the constraints in the end. Dashed lines correspond to objective AUC values (y-axis on the right) and solid lines correspond to constraint violations (y-axis on the left).

where $\mathcal{D}_+/\mathcal{D}_-$ denote the set of positive/negative examples regardless their groups, respectively.

Our proposed penalty method solves the following problem:

$$\min_{\mathbf{w}} F(\mathbf{w}) + \frac{1}{2|\Gamma|} \sum_{\tau \in \Gamma} \beta[h_{\tau}^+(\mathbf{w})]_+ + \beta[h_{\tau}^-(\mathbf{w})]_+. \tag{9}$$

It is notable that the above problem does not directly satisfy the Assumption 5.1. However, the Algorithm 1 with a minor change is still applicable by formulating $[h_{\tau}^+(\mathbf{w})]_+ = f(g(h_{\tau,1}(\mathbf{w})))$, where $f(g) = [g]_+$, $g(h) = |h| - \kappa$ and $h_{\tau,1}(\mathbf{w}) = \frac{1}{n_p^+} \sum_{i=1}^{n_p} \mathbb{I}\{b_i^p = 1\}\sigma(s_{\mathbf{w}}(\mathbf{a}_i^p) - \tau)$. As a result, $f$ is monotonically non-decreasing and convex, $g$ is convex, and $h_{\tau,1}$ is a smooth function, which fits another setting in Hu et al. (2024). The only change to Algorithm 1 is to maintain an estimator of $h_{\tau,1}, \tau \in \Gamma$ and handle $[h_{\tau}^-(\mathbf{w})]_+$ similarly.

**Setting.** For experiments, we use three datasets, Adult and COMPAS used in Donini et al. (2018), CheXpert (Irvin et al., 2019), which contain male/female, Caucasian/non-Caucasian, and male/female groups, respectively. We set $\Gamma = \{-3, -2, -1, \cdots, 3\}$. For models, we use a simple neural network with 2 hidden layers for Adult and COMPAS data and use DenseNet121 for CheXpert data (Yuan et al., 2021). We implement Algorithm 1 with setting $\gamma_2 = 0.8$, $\gamma_2' = 0.1$, $\kappa = 0.005$. We compare with the algorithm in Li et al. (2024a) which optimizes a squared-hinge penalty function. For both algorithms, we tune the initial learning rate in {1e-3, 1e-4} and decay it at 50% and 75% epochs by a factor of 10. We tuned $\beta = \{1, 4, 8, 10, 20, 40, 80, 100\}$ for our hinge exact penalty method and $\beta = \{10, 40, 80, 100, 200, 400, 800, 1000\}$ for squared hinge penalty method. We also compare a double-loop method (ICPPAC) (Boob et al., 2023, Algorithm 4) for Adult and COMPAS, where we tune their $\eta$ in {0.1, 0.01}, $\tau$ in {1, 10, 100}, $\mu$ in {1e-2, 1e-3, 1e-4}, and fix $\theta_t$ to 0.1. We do not report ICPPAC on the large-scale CheXpert data because it is not efficient due to back-propagation on 15 individual constraint functions every iteration. We run each method for total 60 epoches with a batch

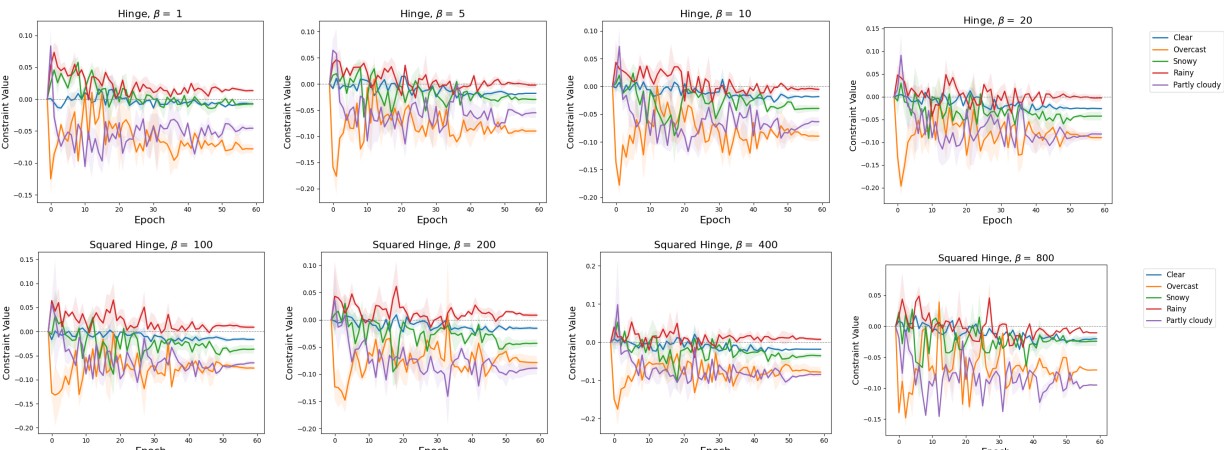

Figure 3: Training curves of 5 constraint values in zero-one loss of different methods for continual learning with non-forgetting constraints when targeting the foggy class. Top: hinge penalty method with different $\beta$; Bottom: squared-hinge penalty method with different $\beta$.

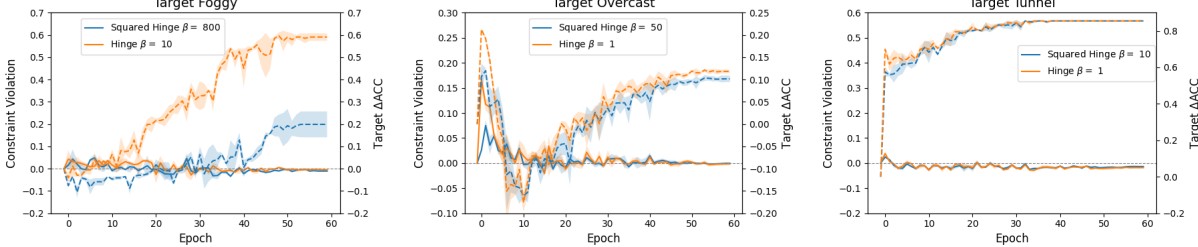

Figure 4: Comparison of training curves between the hinge penalty method and the squared-hinge penalty method when both methods satisfy the constraints in the end. Dashed lines correspond to target accuracy improvement over the base model (y-axis on the right) and solid lines correspond to constraint violation in terms of zero-one loss (y-axis on the left).

size of 128. As optimization involves randomness, we run all the experiments with five different random seeds and then calculate average of the AUC scores and constraint values.

**Result.** We present the training curves of values of constraint functions in Fig. 1 on Adult dataset to illustrate which $\beta$ value satisfies the constraints. For the hinge exact penalty method, when $\beta = 20$, all constraints are satisfied. In contrast, the squared hinge penalty method requires a much larger $\beta = 800$ to satisfy the constraints at the end. We include more results on COMPAS and CheXpert in Appendix A. We compare the training curves of objective AUC values and the constraint violation as measured by the worst constraint function value at each epoch of different methods in Fig. 2 on different datasets. These results demonstrate that the hinge exact penalty method has a better performance than the squared hinge penalty method in terms of the objective AUC value when both have a similar constraint satisfaction, and is competitive with if not better than ICPPAC method.

## 6.2 Continual learning with non-forgetting constraints

Continual learning with non-forgetting constraints has been considered in Li et al. (2024a), which is termed the model developmental safety. We consider the same problem of developing the CLIP model while satisfying developmental safety constraints:

$$\min_{\mathbf{w}} \ F(\mathbf{w}, \mathcal{D}) := \frac{1}{n} \sum_{(\mathbf{x}_i, \mathbf{t}_i) \in \mathcal{D}} L_{\text{ctr}}(\mathbf{w}, \mathbf{x}_i, \mathbf{t}_i, \mathcal{T}_i^-, \mathcal{I}_i^-),$$
$$\text{s.t. } h_k := \mathcal{L}_k(\mathbf{w}, D_k) - \mathcal{L}_k(\mathbf{w}_{\text{old}}, D_k) \leq 0, k = 1, \cdots, m.$$

The $L_{\mathrm{ctr}}(\mathbf{w}, \mathbf{x}_i, \mathbf{t}_i, \mathcal{T}_i^-, \mathcal{I}_i^-)$ is a two-way contrastive loss for each image-text pair $(\mathbf{x}_i, \mathbf{t}_i)$ (Yuan et al., 2022),

$$L_{\mathrm{ctr}}(\mathbf{w}; \mathbf{x}_i, t_i, \mathcal{T}_i^-, \mathcal{I}_i^-) := -\tau \log \frac{\exp(E_1(\mathbf{w}, \mathbf{x}_i)^\top E_2(\mathbf{w}, t_i)/\tau)}{\sum_{t_j \in \mathcal{T}_i^-} \exp(E_1(\mathbf{w}, \mathbf{x}_i)^\top E_2(\mathbf{w}, t_j)/\tau)}$$
$$- \tau \log \frac{\exp(E_2(\mathbf{w}, t_i)^\top E_1(\mathbf{w}, \mathbf{x}_i)/\tau)}{\sum_{\mathbf{x}_j \in \mathcal{I}_i^-} \exp(E_2(\mathbf{w}, t_j)^\top E_1(\mathbf{w}, \mathbf{x}_i)/\tau)},$$

where $E_1(\mathbf{w}, \mathbf{x})$ and $E_2(\mathbf{w}, t)$ denotes a (normalized) encoded representation of an image $\mathbf{x}$ and a text $t$, respectively. $\mathcal{T}_i^-$ denotes the set of all texts to be contrasted with respect to (w.r.t) $\mathbf{x}_i$ (including itself) and $\mathcal{I}_i^-$ denotes the set of all images to be contrasted w.r.t $t_i$ (including itself). Here, the data $\mathcal{D}$ is a target dataset. $\mathcal{L}_k(\mathbf{w}, \mathcal{D}_k) = \frac{1}{n_k} \sum_{(\mathbf{x}_i, y_i) \sim \mathcal{D}_k} \ell_k(\mathbf{w}, \mathbf{x}_i, y_i)$, where $\ell_k$ is a logistic loss for the $k$-th classification task. We use our Algorithm 2 to optimize the hinge penalty function.

**Setting.** We follow the exactly same experiment setting as in Li et al. (2024a). We experiment on the large-scale diverse driving image dataset, namely BDD100K (Yu et al., 2020). This dataset involves classification of six weather conditions, i.e., clear, overcast, snowy, rainy, partly cloudy, foggy, and of six scene types, i.e., highway, residential area, city street, parking lot, gas station, tunnel. We consider three objectives of contrastive loss for finetuning a pretrained CLIP model to target at improving classification of foggy, tunnel and overcast, respectively. For each task, we use other weather or scene types to formulate the constraints, ensuring the new model does not lose the performance on these other classes. The data of the objective for the target class are sampled from the training set of BDD100K and the external LAION400M dataset (Schuhmann et al., 2021) and the data of constrained tasks are sample from BDD100K, where the statistics are given in Li et al. (2024a).

For all methods, we fix the learning rate $1e^{-6}$ with cosine scheduler, using a weight decay of 0.1. We run each method for a total of 60 epochs with a batch size of 256 and 400 iterations per epoch. We tune $\beta$ in $\{0.01, 0.1, 1, 5, 10, 20\}$ for our method and in $\{80, 100, 200, 400, 800\}$ for squared hinge penalty method and set $|\mathcal{B}_c| = m, |\mathcal{B}_k| = 10, \gamma_1 = \gamma_2 = 0.8, \gamma_1' = \gamma_2' = 0.1, \tau = 0.05$.

**Results.** We present the training curves of the constraint functions as measured by zero-one loss in Fig. 3 for different methods with different $\beta$ values on targeting foggy class. We can see that when $\beta = 10$, our hinge penalty method will satisfy all constraints. In contrast, the squared hinge penalty method needs a much larger $\beta = 800$ to satisfy all constraints. The constraint curves when targeting overcast and tunnel are presented in Figs. 5, 6 in Appendix A.

Fig. 4 compares the training curves of the target accuracy improvement between our hinge penalty method and squared-hinge penalty method when both satisfy the constraints. According to the Fig. 4, our hinge penalty method converges faster in terms of the objective measure when they exhibit similar constraint satisfaction in the end.

# 7 Conclusion and Discussion

In this paper, we have studied non-convex constrained optimization with a weakly-convex objective and weakly convex constraint functions, which are all stochastic. We developed a hinge-based exact penalty method and its theory to guarantee finding a nearly $\epsilon$-KKT solution. By leveraging stochastic optimization techniques for non-smooth weakly-convex finite-sum coupled compositional optimization problems, we developed algorithms for solving the penalty function with different structures of the objective functions. Our experiments on fair learning with ROC fairness constraints and continual learning with non-forgetting constraints demonstrate the effectiveness of our algorithms compared. One limitation of this work is that the rate is still worse than non-convex smooth equality constrained optimization. Different from our hinge-based exact penalty method, there are some relevant works on smoothed proximal and primal–dual methods with constant penalty parameters(Pu et al., 2024; Huang et al., 2025), which involve any dual variables or primal–dual updates and make valuable contributions in convex or linearly constrained settings. It is indeed an interesting future direction to explore whether primal–dual techniques can be extended to fully non-convex constrained optimization in the stochastic setting, potentially achieving even better convergence guarantees.

**Acknowledgments**

M. Yang, G. Li, Q. Hu and T. Yang were partially supported by the National Science Foundation Career Award 2246753, the National Science Foundation Award 2246757 and 2306572. Q. Lin was partially supported by National Science Foundation Award 2147253. The authors thank the action editor and the anonymous reviewers for their insightful suggestions and constructive feedback.

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

## A More Experiment Results

The experiments of AUC Maximization with ROC Fairness Constraints in our paper is run on a GPU server with 10 A30 24G GPUs. Each seed in this experiment takes about 4 hours to complete. The experiment of continual learning with non-forgetting constraints is run on two A100 40GB GPUs. Each seed in this setting takes approximately 12 hours to finish. These runtimes are comparable across all methods since they share identical model architectures and computational setups, including the learning-rate schedule, weight decay, batch sizes, and the selections of $|\mathcal{B}_c|$ and $|\mathcal{B}_k|$. We tune the penalty coefficient $\beta$, which allows us to study how different methods reduce constraint violations and how their performances in the objective value change when the constraints are satisfied. This consistent setup provides a fair and controlled comparison of the optimization behavior of all methods.

We give other experiment results in this Appendix. First, we compute the minimum singular value $\delta$ of the Jacobian matrix of violating constraint functions in our experiments at different solutions to verify the conditions in Lemma 4.4, as shown in the Table 2. For the first experiment, we compute $\sigma$ based on the Adult dataset at the initial model, final model, and two randomly selected intermediate epochs. For our second experiment, we compute $\sigma$ when Target Foggy at both the initial and final models. All results demonstrate that the minimum singular value $\sigma$ of $\partial H(\mathbf{x})$ remains positive, which is consistent with the theoretical guarantee stated in Lemma 4.4.

| Experiment 1 (Adult) | Initial Model | Epoch_30 | Epoch_45 | Final Model |
|---|---|---|---|---|
| Minimum Singular Values $\sigma$ of $\partial H(\mathbf{x})$ | 0.000648 | 0.000112 | 0.000102 | 0.000101 |

| Experiment 2 (Target foggy) | Initial Model | Final Model |
|---|---|---|
| Minimum Singular Values $\sigma$ of $\partial H(\mathbf{x})$ | 24.1038 | 16.3397 |

Table 2: Minimum singular values $\sigma$ of the Jacobian matrix $\partial H(\mathbf{x})$ at different solution.

For targeting overcast we consider other weather conditions except for foggy as protected tasks due to that there is a lack of foggy data in BDD100k for defining a significant constraint. For the same reason, we consider other scence types except gas station as protected tasks for targeting tunnel. We use the mostly same experiment setting with the case target foggy only difference when tuning $\beta$. We tune $\beta = \{0.01, 0.1, 1, 10, 20\}$ for hinge-based penalty method and $\beta = \{0.1, 1, 20, 50, 100, 200\}$ for squared hinge penalty method when target overcast. When target tunnel, we tune $\beta = \{0.01, 0.1, 1, 10, 20\}$ for both hinge-based penalty method for squared hinge penalty method. The training curves of target overcast and tunnel show in Fig. 5 and Fig. 6. The training curves on dataset COMPAS and Chexpert show in Fig. 7 and Fig. 8 for experiment ROC Fairness Constraints.

## B Technique Lemmas

To prove our main theorem, we need the following lemmas.

The following lemma follows from standard result (Davis & Drusvyatskiy, 2019).
**Lemma B.1.** *Given a $\rho$ weakly convex function $\phi$ and $\theta < (\rho)^{-1}$, then the envelope $\phi_\theta$ is smooth with gradient given by $\nabla \phi_\theta(\mathbf{x}) = \theta^{-1}(\mathbf{x} - prox_{\theta\phi}(\mathbf{x}))$, where*

$$prox_{\theta\phi}(\mathbf{x}) := \arg\min_{\mathbf{y}} \left\{ \phi(\mathbf{y}) + \frac{1}{2\theta}\|\mathbf{y} - \mathbf{x}\|^2 \right\}. \tag{10}$$

*The smoothness constant of $\phi_\theta$ is $\frac{2-\theta\rho}{\theta(1-\theta\rho)}$. In addition, $dist(0, \partial\phi(\bar{\mathbf{x}})) \leq \|\nabla\phi_\theta(\mathbf{x})\|$.*

**Lemma B.2.** *[Lemma 1 (Jiang et al., 2022)] Consider update (26). Under Assumptions 3.1, 5.1 and D.1, by setting $\gamma_1' = \frac{n-|\mathcal{B}|}{|\mathcal{B}|(1-\gamma_1)} + (1-\gamma_1)$, for $\gamma_1 \leq \frac{1}{2}$, the function value variance $\Xi^{t+1} := \frac{1}{n}\sum_{i=1}^{n}\|u_{1,i}^{t+1} - g_i(\mathbf{x}_{t+1})\|^2$ can be bounded as*

$$\mathbb{E}\left[\Xi^{t+1}\right] \leq (1 - \frac{|\mathcal{B}|\gamma_1}{n})\mathbb{E}\left[\Xi^t\right] + \frac{8nL_g^2}{|\mathcal{B}|}\mathbb{E}[\|\mathbf{x_{t+1}} - \mathbf{x_t}\|^2] + \frac{2\gamma_1^2|\mathcal{B}|\sigma_g^2}{n|\mathcal{B}_{1,i}|}. \tag{11}$$

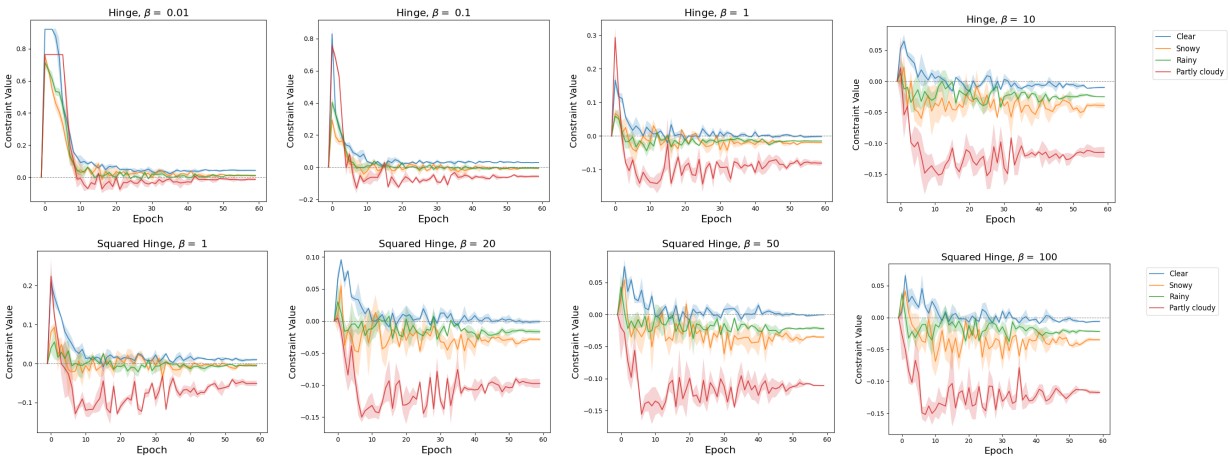

Figure 5: Training curves of 4 constraint values in zero-one loss of different methods for continual learning with non-forgetting constraints when targeting the overcast class. Top: hinge penalty method with different $\beta$; Bottom: squared-hinge penalty method with different $\beta$.

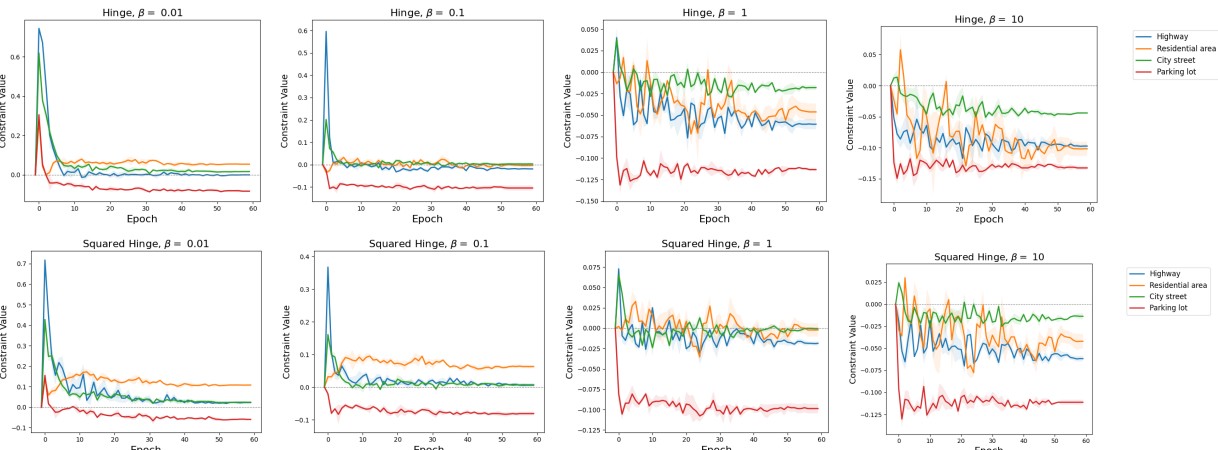

Figure 6: Training curves of 4 constraint values in zero-one loss of different methods for continual learning with non-forgetting constraints when targeting the tunnel class. Top: hinge penalty method with different $\beta$; Bottom: squared-hinge penalty method with different $\beta$.

**Lemma B.3.** *[Lemma 1 (Jiang et al., 2022)] Consider the update Line 6 in Algorithm 1. Under Assumptions 3.1 and 5.1, by setting $\gamma_2' = \frac{m - |\mathcal{B}_c|}{|\mathcal{B}_c|(1-\gamma_2)} + (1 - \gamma_2)$, for $\gamma_2 \leq \frac{1}{2}$, the function value variance $\Gamma^{t+1} := \frac{1}{m} \sum_{k=1}^{m} \|u_{2,k}^{t+1} - h_k(\mathbf{x}_{t+1})\|^2$ can be bounded as*

$$\mathbb{E}\left[\Gamma_{t+1}\right] \leq (1 - \frac{|\mathcal{B}_c|\gamma_2}{m})\mathbb{E}[\Gamma_t] + \frac{8mL_h^2}{|\mathcal{B}_c|}\mathbb{E}[\|\mathbf{x}_{t+1} - \mathbf{x}_t\|^2] + \frac{2\gamma_1^2|\mathcal{B}_c|\sigma_h^2}{m|\mathcal{B}_{2,k}|}. \tag{12}$$

From the analysis of Lemma 4.5 in Hu et al. (2024), the following two inequalities hold true based on Lemma B.2 and Lemma B.3. For simplicity, we use a constant $M^2 \geq \max\{2L_f^2 L_g^2 + 2\beta^2 L_h^2, 2\sigma_f^2 + 2L_F^2 + 2\beta^2 L_h^2\}$.

$$\mathbb{E}[\frac{1}{m}\sum_{k=1}^{m}\|h_k(\mathbf{x}_t) - u_{2,k}^t\|] \leq (1 - \frac{|\mathcal{B}_c|\gamma_2}{2m})^t \frac{1}{m}\sum_{k=1}^{m}\|u_{2,k}^0 - h_k(\mathbf{x}_0)\| + \frac{4mL_h\eta M}{|\mathcal{B}_c|\gamma_2^{1/2}} + \frac{2\gamma_2^{1/2}\sigma_h}{|\mathcal{B}_{2,k}|^{1/2}}, \tag{13}$$

and

$$\mathbb{E}[\frac{1}{n}\sum_{i=1}^{n}\|g_i(\mathbf{x}_t) - u_{1,i}^t\|] \leq (1 - \frac{|\mathcal{B}|\gamma_1}{2n})^t \frac{1}{n}\sum_{i=1}^{n}\|u_{1,i}^0 - g_i(\mathbf{x}_0)\| + \frac{4nL_g M\eta}{|\mathcal{B}|\gamma_1^{1/2}} + \frac{2\gamma_1^{1/2}\sigma_g}{|\mathcal{B}_{1,i}|^{1/2}}. \tag{14}$$

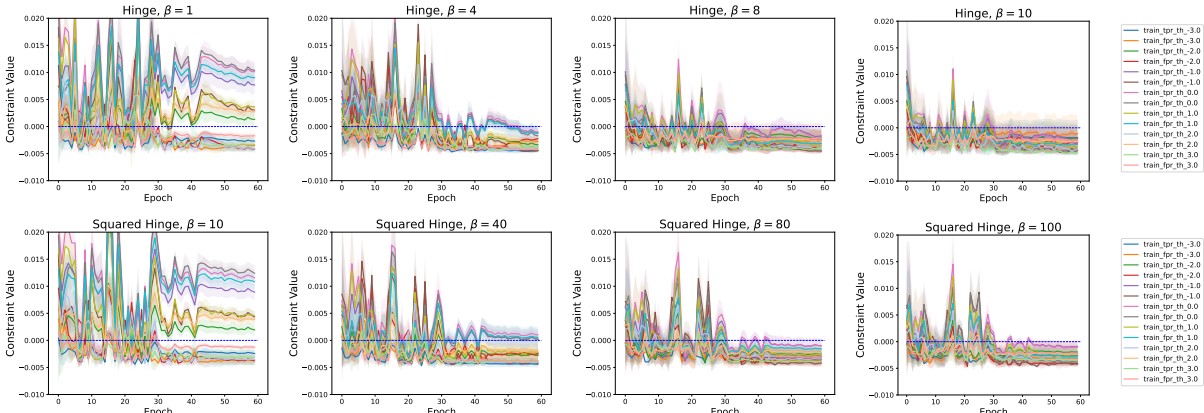

Figure 7: Training curves of 15 constraint functions of different methods for fair learning on the COMPAS dataset. Top: hinge-based penalty method with different $\beta$; Bottom: squared-hinge-based penalty method with different $\beta$.

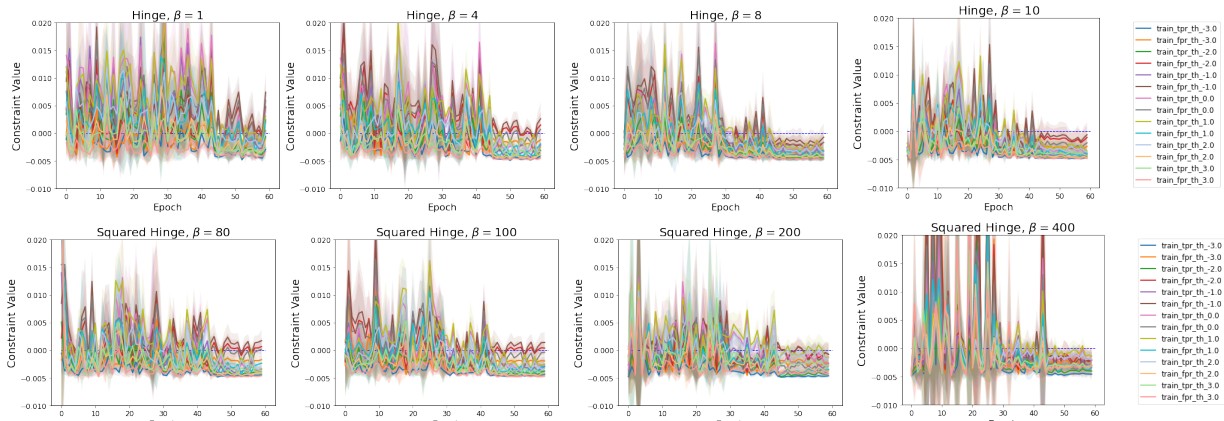

Figure 8: Training curves of 15 constraint functions of different methods for fair learning on the CheXpert dataset. Top: hinge-based penalty method with different $\beta$; Bottom: squared-hinge-based penalty method with different $\beta$.

*Proof of Lemma 4.1.* Notice that we have assumed $F(\cdot)$ to be $\rho_0$-weakly convex in Assumption 3.1. By the definition of weak convexity, it suffices to show $[h_k(\mathbf{x})]_+$ to be $\rho_1$-weakly convex for all $k = 1, \ldots, m$.

Let $[h_k(x)]'_+$ denote any element in $\partial[h_k(x)]_+$. Given any $k \in \{1, \ldots, m\}$ and $x, x'$, following from the convexity of $[\cdot]_+$, we have

$$
\begin{aligned}
[h_k(x)]_+ - [h_k(x')]_+ &\geq \partial[h_k(x')]_+(h(x) - h(x')) \\
&\overset{(a)}{\geq} \partial[h_k(x')]_+(\langle\partial h_k(x'), x - x'\rangle - \frac{\rho_1}{2}\|x - x'\|^2) \\
&\overset{(b)}{\geq} \langle\partial h_k(x')\partial[h_k(x')]_+, x - x'\rangle - \frac{\rho_1}{2}\|x - x'\|^2,
\end{aligned}
\tag{15}
$$

where $(a)$ follows from the fact $[h_k(x')]'_+ \geq 0$ for all $x'$ and the $\rho_1$-weak convexity of $h_k(\cdot)$, and $(b)$ follows from the fact $[h_k(x')]'_+ \leq 1$.

To show the $L$-Lipschitz continuity of $\Phi(\cdot)$, we utilize the $L_F$ and $L_h$-Lipschitz continuity of $F(\cdot)$ and $h_k(\cdot)$ to obtain that for any $x, x'$ we have

$$
\begin{aligned}
\|\Phi(x) - \Phi(x')\| &\leq \|F(x) - F(x')\| + \frac{\beta}{m} \sum_{k=1}^{m} \|[h_k(x)]_+ - [h_k(x')]_+\| \\
&\leq L_F \|x - x'\| + \frac{\beta}{m} \sum_{k=1}^{m} \|h_k(x) - h_k(x')\| \\
&\leq L_F \|x - x'\| + \frac{\beta}{m} \sum_{k=1}^{m} L_h \|x - x'\| \\
&\leq (L_F + \beta L_h) \|x - x'\|.
\end{aligned}
\tag{16}
$$

$\square$

*Proof of Lemma 4.3.* Our goal is to prove that there exists a constant $\delta > 0$ for any $\mathbf{x}$ such that $h(\mathbf{x}) > 0$ (violating the constraint), we have $\operatorname{dist}(0, \partial h(\mathbf{x})) \geq \delta$.

By conditions (i) and (ii) and the fact that $h(x) > 0$, we have

$$
c < h(\mathbf{x}) - \min_{\mathbf{y}} h(\mathbf{y}) \leq \frac{\operatorname{dist}(0, \partial h(\mathbf{x}))^2}{2\mu}.
$$

This means $\|\operatorname{dist}(0, \partial h(\mathbf{x}))\| \geq \sqrt{2c\mu} =: \delta$.

$\square$

*Proof of Lemma 4.4.* Let $\boldsymbol{\xi} = (\xi_1, \ldots, \xi_m)^T$, where $\xi_k$ defined as

$$
\xi_k = \begin{cases} 1 & \text{if } h_k(\mathbf{x}) > 0, \\ [0, 1] & \text{if } h_k(\mathbf{x}) = 0, \\ 0 & \text{if } h_k(\mathbf{x}) < 0, \end{cases} \in [h_k(\mathbf{x})]'_+.
$$

For any $\mathbf{x}$ and any sub-Jacobian $G(\mathbf{x}) \in \partial H(\mathbf{x})$, let $g_k(\mathbf{x})^\top$ denote its $k$-th row (so $g_k(\mathbf{x}) \in \partial h_k(\mathbf{x})$).

$$
\left\{ \sum_{k=1}^{m} \xi_k g_k(\mathbf{x}) \right\} = \{G(\mathbf{x})^\top \boldsymbol{\xi}\} \subset \sum_{k=1}^{m} \partial [h_k(\mathbf{x})]_+.
$$

Let $\mathcal{V} := \{\mathbf{x} : \max_k h_k(\mathbf{x}) > 0\}$. For any $\mathbf{x} \in \mathcal{V}$ we have

$$
\operatorname{dist}\left(0, \frac{1}{m} \sum_{k=1}^{m} \partial [h_k(\mathbf{x})]_+\right) \geq \left\| \frac{1}{m} G(\mathbf{x})^\top \boldsymbol{\xi} \right\| \geq \frac{1}{m} \lambda_{\min}(G(\mathbf{x})) \|\boldsymbol{\xi}\| \geq \frac{1}{m} \lambda_{\min}(G(\mathbf{x})).
$$

By the assumption that $\operatorname{dist}(0, \lambda_{\min}(\partial H(\mathbf{x}))) \geq \sigma > 0, \forall \mathbf{x}$ such that $\max_k h_k(\mathbf{x}) > 0$, we have

$$
\lambda_{\min}(G(\mathbf{x})) \geq \sigma \quad \forall G(\mathbf{x}) \in \partial H(\mathbf{x}).
$$

Therefore, for all $\mathbf{x} \in \mathcal{V}$,

$$
\operatorname{dist}\left(0, \frac{1}{m} \sum_{k=1}^{m} \partial [h_k(\mathbf{x})]_+\right) \geq \frac{\sigma}{m}
$$

$\square$

## C  Proof of Theorem 5.3

*Proof.* The objective function

$$
\min_{\mathbf{x}} \Phi(\mathbf{x}) := \underbrace{\mathbb{E}_\zeta[f(\mathbf{x}; \zeta)]}_{F(\mathbf{x})} + \underbrace{\frac{\beta}{m} \sum_{k=1}^{m} [h_k(\mathbf{x})]_+}_{H(\mathbf{x})}.
\tag{17}
$$

For simplicity, denote $\bar{\mathbf{x}}_t := \text{prox}_{\Phi/\bar{\rho}}(\mathbf{x}_t)$. Consider change in the Moreau Envelope

$$
\begin{aligned}
\mathbb{E}_t[\Phi_{1/\bar{\rho}}(\mathbf{x}_{t+1})] &= \mathbb{E}_t[\min_{\tilde{\mathbf{x}}} \Phi(\tilde{\mathbf{x}}) + \frac{\bar{\rho}}{2}\|\tilde{\mathbf{x}} - \mathbf{x}_{t+1}\|^2] \\
&\leq \mathbb{E}_t[\Phi(\bar{\mathbf{x}}_t) + \frac{\bar{\rho}}{2}\|\bar{\mathbf{x}}_t - \mathbf{x}_{t+1}\|^2] \\
&= \mathbb{E}_t[\Phi(\bar{\mathbf{x}}_t) + \frac{\bar{\rho}}{2}\|\bar{\mathbf{x}}_t - (\mathbf{x}_t - \eta(G_1^t + G_2^t))\|^2] \\
&\leq \Phi(\bar{\mathbf{x}}_t) + \frac{\bar{\rho}}{2}\|\bar{\mathbf{x}}_t - \mathbf{x}_t\|^2 + \bar{\rho}\mathbb{E}_t[\eta\langle\bar{\mathbf{x}}_t - \mathbf{x}_t, G_1^t + G_2^t\rangle] + \eta^2\frac{\bar{\rho}M^2}{2} \\
&= \Phi_{1/\bar{\rho}}(\mathbf{x}_t) + \bar{\rho}\eta\langle\bar{\mathbf{x}}_t - \mathbf{x}_t, \mathbb{E}_t[G_1^t]\rangle + \bar{\rho}\eta\langle\bar{\mathbf{x}}_t - \mathbf{x}_t, \mathbb{E}_t[G_2^t]\rangle + \eta^2\frac{\bar{\rho}M^2}{2}, \quad (18)
\end{aligned}
$$

where the second inequality uses the bound of $\mathbb{E}_t[\|G_1^t + G_2^t\|^2]$, which follows from the Lipchitz continuity and bounded variance, denoted by $M^2 \geq 2\sigma_f^2 + 2L_F^2 + 2\beta^2 L_h^2$. Here,

$$
\mathbb{E}_t[G_1^t] \in \partial F(\mathbf{x}_t), \quad \mathbb{E}_t[G_2^t] \in \frac{\beta}{m}\sum_{k=1}^m \partial h_k(\mathbf{x}_t)[u_{2,k}^t]_+'.
$$

Next we give the bound of $\langle\bar{\mathbf{x}}_t - \mathbf{x}_t, \mathbb{E}_t[G_1^t]\rangle$ and $\langle\bar{\mathbf{x}}_t - \mathbf{x}_t, \mathbb{E}_t[G_2^t]\rangle$. To this end, first we give the bound of $\langle\bar{\mathbf{x}}_t - \mathbf{x}_t, \mathbb{E}_t[G_1^t]\rangle$. Since $F$ is $\rho_0$-weakly convex, we have

$$
F(\bar{\mathbf{x}}_t) - F(\mathbf{x}_t) \geq \partial F(\mathbf{x}_t)^\top(\bar{\mathbf{x}}_t - \mathbf{x}_t) - \frac{\rho_0}{2}\|\bar{\mathbf{x}}_t - \mathbf{x}_t\|^2.
$$

Then it follows

$$
\partial F(\mathbf{x}_t)^\top(\bar{\mathbf{x}}_t - \mathbf{x}_t) \leq F(\bar{\mathbf{x}}_t) - F(\mathbf{x}_t) + \frac{\rho_0}{2}\|\bar{\mathbf{x}}_t - \mathbf{x}_t\|^2. \quad (19)
$$

Next we bound $\langle\bar{\mathbf{x}}_t - \mathbf{x}_t, \mathbb{E}_t[G_2^t]\rangle$. For given $k \in \{1, \ldots, m\}$, we get

$$
\begin{aligned}
[h_k(\bar{\mathbf{x}}_t)]_+ - [u_{2,k}^t]_+ &\geq [u_{2,k}^t]_+'(h_k(\bar{\mathbf{x}}_t) - u_{2,k}^t) \\
&\geq [u_{2,k}^t]_+'\left[h_k(\mathbf{x}_t) - u_{2,k}^t + \partial h_k(\mathbf{x}_t)^\top(\bar{\mathbf{x}}_t - \mathbf{x}_t) - \frac{\rho_1}{2}\|\bar{\mathbf{x}}_t - \mathbf{x}_t\|^2\right] \\
&\geq [u_{2,k}^t]_+'(h_k(\mathbf{x}_t) - u_{2,k}^t) + [u_{2,k}^t]_+'\partial h_k(\mathbf{x}_t)^\top(\bar{\mathbf{x}}_t - \mathbf{x}_t) - \frac{\rho_1}{2}\|\bar{\mathbf{x}}_t - \mathbf{x}_t\|^2,
\end{aligned}
$$

where the first inequality uses the convexity of function $[\cdot]_+$, the second inequality uses the fact $[u_{2,k}^t]_+' \geq 0$ and the $\rho_1$-weak convexity of $h_k(\cdot)$, and the last inequality uses the fact that $0 \leq [u_{2,k}^t]_+' \leq 1$. Then it follows

$$
\begin{aligned}
&\beta\frac{1}{m}\sum_{k=1}^m [u_{2,k}^t]_+'\partial h_k(\mathbf{x}_t)^\top(\bar{\mathbf{x}}_t - \mathbf{x}_t) \\
&\leq \frac{\beta}{m}\sum_{k=1}^m\left[[h_k(\bar{\mathbf{x}}_t)]_+ - [u_{2,k}^t]_+ - [u_{2,k}^t]_+'(h_k(\mathbf{x}_t) - u_{2,k}^t) + \frac{\rho_1}{2}\|\bar{\mathbf{x}}_t - \mathbf{x}_t\|^2\right]. \quad (20)
\end{aligned}
$$

Adding above two estimation equation 19 and equation 20 back to equation 18, we have

$$
\begin{aligned}
\mathbb{E}_t[\Phi_{1/\bar{\rho}}(\mathbf{x}_{t+1})] &\leq \Phi_{1/\bar{\rho}}(\mathbf{x}_t) + \bar{\rho}\eta\langle\bar{\mathbf{x}}_t - \mathbf{x}_t, \mathbb{E}_t[G_1^t]\rangle + \bar{\rho}\eta\langle\bar{\mathbf{x}}_t - \mathbf{x}_t, \mathbb{E}_t[G_2^t]\rangle + \eta^2\frac{\bar{\rho}M^2}{2} \\
&\leq \Phi_{1/\bar{\rho}}(\mathbf{x}_t) + \eta^2\frac{\bar{\rho}M^2}{2} + \bar{\rho}\eta\left[F(\bar{\mathbf{x}}_t) - F(\mathbf{x}_t) + \frac{\rho_0}{2}\|\bar{\mathbf{x}}_t - \mathbf{x}_t\|^2\right] \\
&\quad + \bar{\rho}\eta\frac{\beta}{m}\sum_{k=1}^m\left[[h_k(\bar{\mathbf{x}}_t)]_+ - [u_{2,k}^t]_+ - [u_{2,k}^t]_+'(h_k(\mathbf{x}_t) - u_{2,k}^t) + \frac{\rho_1}{2}\|\bar{\mathbf{x}}_t - \mathbf{x}_t\|^2\right] \\
&= \Phi_{1/\bar{\rho}}(\mathbf{x}_t) + \eta^2\frac{\bar{\rho}M^2}{2} + \bar{\rho}\eta\left[F(\bar{\mathbf{x}}_t) - F(\mathbf{x}_t) + \frac{\rho_0}{2}\|\bar{\mathbf{x}}_t - \mathbf{x}_t\|^2\right] \\
&\quad + \bar{\rho}\eta\frac{\beta}{m}\sum_{k=1}^m\left[[h_k(\bar{\mathbf{x}}_t)]_+ - [h_k(\mathbf{x}_t)]_+ + [h_k(\mathbf{x}_t)]_+ - [u_{2,k}^t]_+ - [u_{2,k}^t]_+'(h_k(\mathbf{x}_t) - u_{2,k}^t) + \frac{\rho_1}{2}\|\bar{\mathbf{x}}_t - \mathbf{x}_t\|^2\right].
\end{aligned}
$$
$$(21)$$

By Lemma 4.1, the function $\Phi$ is $C$-weakly convex. We have $(\bar{\rho} - C)-$strong convexity of $\mathbf{x} \mapsto \Phi(\mathbf{x}) + \frac{\bar{\rho}}{2}\|\mathbf{x}_t - \mathbf{x}\|^2$

$$\Phi(\bar{\mathbf{x}}_t) - \Phi(\mathbf{x}_t) = \left(\Phi(\bar{\mathbf{x}}_t) + \frac{\bar{\rho}}{2}\|\mathbf{x}_t - \bar{\mathbf{x}}_t\|^2\right) - \left(\Phi(\mathbf{x}_t) + \frac{\bar{\rho}}{2}\|\mathbf{x}_t - \mathbf{x}_t\|^2\right) - \frac{\bar{\rho}}{2}\|\bar{\mathbf{x}}_t - \mathbf{x}_t\|^2$$

$$\leq (\frac{C}{2} - \bar{\rho})\|\bar{\mathbf{x}}_t - \mathbf{x}_t\|^2.$$

Then we get

$$F(\bar{\mathbf{x}}_t) - F(\mathbf{x}_t) + \beta\frac{1}{m}\sum_{k=1}^m ([h_k(\bar{\mathbf{x}}_t)]_+ - [h_k(\mathbf{x}_t)]_+) = \Phi(\bar{\mathbf{x}}_t) - \Phi(\mathbf{x}_t) \leq (\frac{C}{2} - \bar{\rho})\|\bar{\mathbf{x}}_t - \mathbf{x}_t\|^2. \tag{22}$$

Plugging inequality equation 22 to equation 21, we get

$$\mathbb{E}_t[\Phi_{1/\bar{\rho}}(\mathbf{x}_{t+1})]$$

$$\leq \Phi_{1/\bar{\rho}}(\mathbf{x}_t) + \eta^2\frac{\bar{\rho}M^2}{2} + \bar{\rho}\eta\Big(\frac{\rho_0}{2} + \beta\frac{\rho_1}{2} + \frac{C}{2} - \bar{\rho}\Big)\|\bar{\mathbf{x}}_t - \mathbf{x}_t\|^2$$

$$+ \bar{\rho}\eta\frac{\beta}{m}\sum_{k=1}^m \Big[[h_k(\mathbf{x}_t)]_+ - [u_{2,k}^t]_+ - [u_{2,k}^t]_+'(h_k(\mathbf{x}_t) - u_{2,k}^t)\Big].$$

Setting $\frac{\rho_0}{2} + \beta\frac{\rho_1}{2} + \frac{C}{2} = \frac{\bar{\rho}}{2}$, we have

$$\mathbb{E}_t[\Phi_{1/\bar{\rho}}(\mathbf{x}_{t+1})]$$

$$\leq \Phi_{1/\bar{\rho}}(\mathbf{x}_t) + \eta^2\frac{\bar{\rho}M^2}{2} - \eta\frac{\bar{\rho}^2}{2}\|\bar{\mathbf{x}}_t - \mathbf{x}_t\|^2 + \bar{\rho}\eta\frac{\beta}{m}\sum_{k=1}^m \Big[[h_k(\mathbf{x}_t)]_+ - [u_{2,k}^t]_+ - [u_{2,k}^t]_+'(h_k(\mathbf{x}_t) - u_{2,k}^t)\Big]$$

$$\leq \Phi_{1/\bar{\rho}}(\mathbf{x}_t) + \eta^2\frac{\bar{\rho}M^2}{2} - \frac{\eta}{2}\|\nabla\Phi_{1/\bar{\rho}}(\mathbf{x}_t)\|^2 + \bar{\rho}\eta\frac{\beta}{m}\sum_{k=1}^m \Big[[h_k(\mathbf{x}_t)]_+ - [u_{2,k}^t]_+ - [u_{2,k}^t]_+'(h_k(\mathbf{x}_t) - u_{2,k}^t)\Big].$$

Since $0 \leq [u_{2,k}^t]_+' \leq 1$, we get

$$\mathbb{E}_t[\Phi_{1/\bar{\rho}}(\mathbf{x}_{t+1})] \leq \Phi_{1/\bar{\rho}}(\mathbf{x}_t) + \eta^2\frac{\bar{\rho}M^2}{2} - \frac{\eta}{2}\|\nabla\Phi_{1/\bar{\rho}}(\mathbf{x}_t)\|^2 + 2\bar{\rho}\eta\frac{\beta}{m}\sum_{k=1}^m \|h_k(\mathbf{x}_t) - u_{2,k}^t\|. \tag{23}$$

Taking the full expectation on both sides of equation 23 and applying the inequality of equation 13, we have

$$\mathbb{E}[\Phi_{1/\bar{\rho}}(\mathbf{x}_{t+1})]$$

$$\leq \mathbb{E}[\Phi_{1/\bar{\rho}}(\mathbf{x}_t)] + \eta^2\frac{\bar{\rho}M^2}{2} - \frac{\eta}{2}\mathbb{E}[\|\nabla\Phi_{1/\bar{\rho}}(\mathbf{x}_t)\|^2]$$

$$+ 2\bar{\rho}\eta\beta\Big[(1 - \frac{|\mathcal{B}_c|\gamma_2}{2m})^t\frac{1}{m}\sum_{k=1}^m \|u_{2,k}^0 - h_k(\mathbf{x}_0)\| + \frac{4mL_h\eta M}{|\mathcal{B}_c|\gamma_2^{1/2}} + \frac{2\gamma_2^{1/2}\sigma_h}{|\mathcal{B}_{2,k}|^{1/2}}\Big].$$

Taking summation from $t = 0$ to $T - 1$ yields

$$\mathbb{E}[\Phi_{1/\bar{\rho}}(\mathbf{x}_T)] \leq \Phi_{1/\bar{\rho}}(\mathbf{x}_0) + \eta^2 T\frac{\bar{\rho}M^2}{2} - \frac{\eta}{2}\sum_{t=0}^{T-1}\mathbb{E}[\|\nabla\Phi_{1/\bar{\rho}}(\mathbf{x}_t)\|^2]$$

$$+ 2\bar{\rho}\eta\beta\Big[\frac{2m}{|\mathcal{B}_c|\gamma_2}\frac{1}{m}\sum_{k=1}^m \|u_{2,k}^0 - h_k(\mathbf{x}_0)\| + T\frac{8mL_h\eta M}{|\mathcal{B}_c|\gamma_2^{1/2}} + T\frac{4\gamma_2^{1/2}\sigma_h}{|\mathcal{B}_{2,k}|^{1/2}}\Big],$$

where we use the fact that $\sum_{t=0}^{T-1}(1 - \mu)^t \leq \frac{1}{\mu}$ for all $\mu \in [0, 1]$. Lower bounding the left-hand-side by $\min_\mathbf{x} \Phi(\mathbf{x})$ and dividing both sides by $T$, we obtain

$$\frac{1}{T}\sum_{t=0}^{T-1}\mathbb{E}[\|\nabla\Phi_{1/\bar{\rho}}(\mathbf{x}_t)\|^2]$$

$$\leq \frac{2}{\eta T}[\Phi_{1/\bar{\rho}}(\mathbf{x}_0) - \min_\mathbf{x}\Phi(\mathbf{x})] + \eta\bar{\rho}M^2 + \frac{C}{T}\frac{m\beta\bar{\rho}}{\gamma_2|\mathcal{B}_c|} + C\Big(\frac{m\bar{\rho}\beta\eta M}{|\mathcal{B}_c|\gamma_2^{1/2}} + \frac{\beta\bar{\rho}\gamma_2^{1/2}}{|\mathcal{B}_{2,k}|^{1/2}}\Big),$$

where $\mathcal{C} = \max\{\frac{8}{m} \sum_{k=1}^{m} \|u_{2,k}^0 - h_k(\mathbf{x}_0)\|, 32L_h, 16\sigma_h\}$. As we set $M^2 \geq 2\sigma_f^2 + 2L_F^2 + 2\beta^2 L_h^2$ and $\frac{\rho_0}{2} + \beta \frac{\rho_1}{2} + \frac{C}{2} = \frac{\bar{\rho}}{2}$, with $\gamma_2 = \mathcal{O}(\frac{|\mathcal{B}_{2,k}|\epsilon^4}{\beta^4})$, $\eta = \mathcal{O}(\frac{|\mathcal{B}_c||\mathcal{B}_{2,k}|^{1/2}\epsilon^4}{\beta^5 m})$ and $T = \mathcal{O}(\frac{\beta^6 m}{|\mathcal{B}_c||\mathcal{B}_{2,k}|^{1/2}\epsilon^6})$, we have

$$\frac{1}{T} \sum_{t=0}^{T-1} \mathbb{E}[\|\nabla\Phi_{1/\bar{\rho}}(\mathbf{x}_t)\|^2] \leq \mathcal{O}(\epsilon^2).$$

By Jensen's inequality, we can get $\mathbb{E}[\|\nabla\Phi_{1/\bar{\rho}}(\mathbf{x}_{\hat{t}})\|] \leq \mathcal{O}(\epsilon)$, where $\hat{t}$ is selected uniformly at random from $\{1, \ldots, T\}$.

$\square$

Now, we will prove that $\mathbf{x}_{\hat{t}}$ is a nearly $\epsilon$-KKT solution, where (i) holds in expectation, (ii) and (iii) in Definition 3.3 hold in high probability, to the original problem (1).

*Proof.* Proof of Corollary 5.4. Let $\bar{\mathbf{x}}_{\hat{t}} = \text{prox}_{1/\bar{\rho}\Phi}(\mathbf{x}_{\hat{t}})$. By optimality of $\bar{\mathbf{x}}_{\hat{t}}$, we have

$$0 \in \partial F(\bar{\mathbf{x}}_{\hat{t}}) + \frac{\beta}{m} \sum_{k=1}^{m} \partial h_k^+(\bar{\mathbf{x}}_{\hat{t}}) + (\bar{\mathbf{x}}_{\hat{t}} - \mathbf{x}_{\hat{t}})\bar{\rho}.$$

Since $\mathbb{E}[\|\nabla\Phi_{1/\bar{\rho}}(\mathbf{x}_{\hat{t}})\|] \leq \mathcal{O}(\epsilon)$, then we have $\mathbb{E}[\|\bar{\mathbf{x}}_{\hat{t}} - \mathbf{x}_{\hat{t}}\|] = \mathbb{E}[\|\nabla\Phi_{1/\bar{\rho}}(\mathbf{x}_{\hat{t}})\|]/\bar{\rho} \leq \epsilon/\bar{\rho}$. Then we have

$$\mathbb{E}\Big[\text{dist}\big(0, \partial F(\bar{\mathbf{x}}_{\hat{t}}) + \frac{\beta}{m} \sum_{k=1}^{m} \partial h_k^+(\bar{\mathbf{x}}_{\hat{t}})\big)\Big] \leq \mathbb{E}[\|(\bar{\mathbf{x}}_{\hat{t}} - \mathbf{x}_{\hat{t}})\bar{\rho}\|] \leq \epsilon.$$

Since $\partial h_k^+(\bar{\mathbf{x}}_{\hat{t}}) = \xi_k \partial h_k(\bar{\mathbf{x}}_{\hat{t}})$ (see (Bauschke et al., 2017, Corollary 16.72)), where

$$\xi_k = \begin{cases} 1 & \text{if } h_k(\bar{\mathbf{x}}_{\hat{t}}) > 0, \\ [0,1] & \text{if } h_k(\bar{\mathbf{x}}_{\hat{t}}) = 0, \quad \in \partial[h_k(\bar{\mathbf{x}}_{\hat{t}})]_+, \\ 0 & \text{if } h_k(\bar{\mathbf{x}}_{\hat{t}}) < 0, \end{cases}$$

there exists $\lambda_k \in \frac{\beta\xi_k}{m} \geq 0, \forall k$ such that $\mathbb{E}[\text{dist}(0, \partial F(\bar{\mathbf{x}}_{\hat{t}}) + \sum_{k=1}^{m} \lambda_k \partial h_k(\bar{\mathbf{x}}_{\hat{t}}))] \leq \epsilon$. Thus, we prove condition (i) in Definition 3.3. Next, let us prove condition (ii) holds with probability $1 - \mathcal{O}(\epsilon)$. Since $\exists \mathbf{v} \in \partial F(\bar{\mathbf{x}}_{\hat{t}})$, under equation (3), we have

$$\text{dist}\left(0, \mathbf{v} + \frac{\beta}{m} \sum_{k=1}^{m} \partial h_k^+(\bar{\mathbf{x}}_{\hat{t}})\right) \geq \text{dist}\left(0, \frac{\beta}{m} \sum_{k=1}^{m} \partial h_k^+(\bar{\mathbf{x}}_{\hat{t}})\right) - \|\mathbf{v}\| \geq \beta\delta - L_F \geq 0. \tag{24}$$

Therefore,

$$\begin{aligned}
\epsilon &\geq \mathbb{E}\Big[\text{dist}\big(0, \mathbf{v} + \frac{\beta}{m} \sum_{k=1}^{m} \partial h_k^+(\bar{\mathbf{x}}_{\hat{t}})\big)\Big] \\
&= \mathbb{E}\Big[\text{dist}\big(0, \mathbf{v} + \frac{\beta}{m} \sum_{k=1}^{m} \partial h_k^+(\bar{\mathbf{x}}_{\hat{t}})\big)\big| \max_k h_k(\bar{\mathbf{x}}_{\hat{t}}) > \epsilon\Big] \text{Prob}(\max_k h_k(\bar{\mathbf{x}}_{\hat{t}}) > \epsilon) \\
&\quad + \mathbb{E}\Big[\text{dist}\big(0, \mathbf{v} + \frac{\beta}{m} \sum_{k=1}^{m} \partial h_k^+(\bar{\mathbf{x}}_{\hat{t}})\big)\big| \max_k h_k(\bar{\mathbf{x}}_{\hat{t}}) \leq \epsilon\Big] \text{Prob}(\max_k h_k(\bar{\mathbf{x}}_{\hat{t}}) \leq \epsilon) \\
&\geq \text{Prob}(\max_k h_k(\bar{\mathbf{x}}_{\hat{t}}) > \epsilon)(\beta\delta - L_F).
\end{aligned}$$

As a result,

$$\text{Prob}(\max_k h_k(\bar{\mathbf{x}}_{\hat{t}}) > \epsilon) \leq \frac{\epsilon}{\beta\delta - L_F} = \mathcal{O}(\epsilon) \tag{25}$$

Thus, it holds with probability $1 - \mathcal{O}(\epsilon)$ that $\max_k h_k(\bar{\mathbf{x}}_{\hat{t}}) \leq \epsilon$. When $\max_k h_k(\bar{\mathbf{x}}_{\hat{t}}) \leq \epsilon$, we have $\lambda_k h_k(\bar{\mathbf{x}}_{\hat{t}}) \leq \lambda_k \max_k h_k(\bar{\mathbf{x}}_{\hat{t}}) \leq \mathcal{O}(\epsilon)$ as $\lambda_k \in \frac{\beta\xi_k}{m}$. It then follows from equation 25 that $\text{Prob}(\lambda_k h_k(\bar{\mathbf{x}}_{\hat{t}}) \geq \mathcal{O}(\epsilon)) \leq \mathcal{O}(\epsilon)$. This proves that $\lambda_k h_k(\bar{\mathbf{x}}_{\hat{t}}) \leq \mathcal{O}(\epsilon)$ with probability $1 - \mathcal{O}(\epsilon)$. If $|h_k(\bar{\mathbf{x}}_{\hat{t}})| < +\infty$ for any $k$ and $\bar{\mathbf{x}}_{\hat{t}}$, this high probability result can be replaced by $\mathbb{E}[\max_k h_k(\bar{\mathbf{x}}_{\hat{t}})] \leq O(\epsilon)$ and $\mathbb{E}[\lambda_k h_k(\bar{\mathbf{x}}_{\hat{t}})] \leq O(\epsilon)$. $\square$

---

**Algorithm 2** *Algorithm for solving equation 2 under Setting II*

---
1: **Initialization:** choose $\mathbf{x}_0, \beta, \gamma_1, \gamma_2$ and $\eta$.
2: **for** $t = 0$ to $T - 1$ **do**
3:   Sample $\mathcal{B}^t \subset \{1, \ldots, n\}$, $\mathcal{B}_c^t \subset \{1, \ldots, m\}$
4:   **for** each $i \in \mathcal{B}^t$ **do**
5:     Sample a mini-batch $\mathcal{B}_{1,i}^t$
6:     Update $u_{1,i}^t$ by $u_{1,i}^{t+1} = (1 - \gamma_1)u_{1,i}^t + \gamma_1 g_i(\mathbf{x}_t; \mathcal{B}_{1,i}^t) + \gamma_1'(g_i(\mathbf{x}_t; \mathcal{B}_{1,i}^t) - g_i(\mathbf{x}_{t-1}; \mathcal{B}_{1,i}^t))$,
7:   **end for**
8:   Let $u_{1,i}^{t+1} = u_{1,i}^t, i \notin \mathcal{B}^t$
9:   Compute $G_1^t = \frac{1}{|\mathcal{B}^t|} \sum_{i \in \mathcal{B}^t} \partial g_i(\mathbf{x}_t; \mathcal{B}_{1,i}^t) \partial f_i(u_{1,i}^t)$.
10:   **for** each $k \in \mathcal{B}_c^t$ **do**
11:     Sample a mini-batch $\mathcal{B}_{2,k}^t$
12:     Update $u_{2,k}^{t+1} = (1 - \gamma_2)u_{2,k}^t + \gamma_2 h_k(\mathbf{x}_t; \mathcal{B}_{2,k}^t) + \gamma_2'(h_k(\mathbf{x}_t; \mathcal{B}_{2,k}^t) - h_k(\mathbf{x}_{t-1}; \mathcal{B}_{2,k}^t))$
13:   **end for**
14:   Let $u_{2,k}^{t+1} = u_{2,k}^t, k \notin \mathcal{B}_c^t$
15:   Compute $G_2^t = \frac{\beta}{|\mathcal{B}_c^t|} \sum_{k \in \mathcal{B}_c^t} \partial h_k(\mathbf{x}_t; \mathcal{B}_{2,k}^t)[u_{2,k}^t]_+'$, $G_1^t = \frac{1}{|\mathcal{B}^t|} \sum_{\zeta \in \mathcal{B}^t} \partial f(\mathbf{x}_t; \zeta)$.
16:   Update $\mathbf{x}_{t+1} = \mathbf{x}_t - \eta(G_1^t + G_2^t)$
17: **end for**

---

# D   Analysis of Setting II

Now, let us consider the second setting where $F(\mathbf{x}) = \frac{1}{n} \sum_{i=1}^n f_i(\mathbb{E}_\zeta[g_i(\mathbf{x}, \zeta)])$. Indeed, this objective is of the same form of the penalty function, which is a coupled compositional function. Let $g_i(\mathbf{x}) = \mathbb{E}_\zeta[g_i(\mathbf{x}, \zeta)]$. We make the following assumption regarding $f_i, g_i$.

**Assumption D.1.** Assume $f_i$ is deterministic and $L_f$-Lipschitz continuous. For any $\mathbf{x}, \mathbf{y}$, $\mathbb{E}_\zeta[|g_i(\mathbf{x}, \zeta) - g_i(\mathbf{y}, \zeta)|^2] \leq L_g^2 \|\mathbf{x} - \mathbf{y}\|^2$, $\mathbb{E}[|g_i(\mathbf{x}, \zeta) - g_i(\mathbf{x})|^2] \leq \sigma_g^2$, and either of the following conditions hold: (i)$f_i$ is monotonically non-decreasing and $\rho_f$-weakly convex, $g_i(\mathbf{x})$ is $\rho_g$-weakly convex; (ii)$f_i$ is $L_{\nabla f}$-smooth, $g_i(\mathbf{x})$ is $L_{\nabla g}$-smooth.

To tackle this objective, we also need to maintain and update estimators for inner functions $g_i(\mathbf{x})$. At the $t$-th iteration, we randomly sample an outer mini-batch $\mathcal{B}^t \in \{1, \ldots, n\}$, and draw inner mini-batch samples $\mathcal{B}_{1,i}^t$ for each $i \in \mathcal{B}^t$ to construct unbiased estimations $g_i(\mathbf{x}_t; \mathcal{B}_{1,i}^t)$. The variance-reduced estimator of $g_i(\mathbf{x}_t)$ denoted by $u_{1,i}^t$ is updated by:

$$u_{1,i}^{t+1} = (1 - \gamma_1)u_{1,i}^t + \gamma_1 g_i(\mathbf{x}_t; \mathcal{B}_{1,i}^t) + \gamma_1'(g_i(\mathbf{x}_t; \mathcal{B}_{1,i}^t) - g_i(\mathbf{x}_{t-1}; \mathcal{B}_{1,i}^t)), \tag{26}$$

where $\gamma_1 \in (0, 1), \gamma_1' = \frac{n - |\mathcal{B}|}{|\mathcal{B}|(1 - \gamma_1)} + 1 - \gamma_1$, where $|\mathcal{B}| = |\mathcal{B}^t|$. Then, we approximate the gradient of $F(\mathbf{x}_t)$ by:

$$G_1^t = \frac{1}{|\mathcal{B}^t|} \sum_{i \in \mathcal{B}^t} \partial g_i(\mathbf{x}_t; \mathcal{B}_{1,i}^t) \partial f_i(u_{1,i}^t). \tag{27}$$

Then we update $\mathbf{x}_{t+1}$ similarly as before by $\mathbf{x}_{t+1} = \mathbf{x}_t - \eta(G_1^t + G_2^t)$. Due to the limit of space, we present full steps in Algorithm 2.

The convergence of Algorithm 2 under condition (i) of Assumption D.1 is stated in Theorem D.2, and that under condition (ii) of Assumption D.1 is stated in Theorem D.3.

**Theorem D.2.** *Suppose Assumption 3.1, 5.1, D.1 with condition (i) hold. Let* $\gamma_1 = \gamma_2 = \mathcal{O}(\min\{|\mathcal{B}_{1,i}|, \frac{|\mathcal{B}_{2,k}|}{\beta^2}\} \frac{\epsilon^4}{\beta^2})$, $\eta = \mathcal{O}(\min\{\frac{|\mathcal{B}|}{n}, \frac{|\mathcal{B}_c|}{\beta m}\} \min\{|\mathcal{B}_{1,i}|^{1/2}, \frac{|\mathcal{B}_{2,k}|^{1/2}}{\beta}\} \frac{\epsilon^4}{\beta^3})$. *After* $T = \mathcal{O}(\max\{\frac{\beta}{|\mathcal{B}_{1,i}|^{1/2}}, \frac{\beta^2}{|\mathcal{B}_{2,k}|^{1/2}}, \frac{1}{|\mathcal{B}_{1,i}|}\} \max\{\frac{n}{|\mathcal{B}|}, \frac{\beta m}{|\mathcal{B}_c|}\} \frac{\beta^3}{\epsilon^6})$ *iterations, the Algorithm 2 satisfies* $\mathbb{E}[\|\nabla \Phi_\theta(\mathbf{x}_{\hat{t}})\|] \leq \epsilon$ *from some* $\theta = O(1/(\rho_0 + \rho_1 \beta))$, *where* $\hat{t}$ *is selected uniformly at random from* $\{1, \ldots, T\}$

**Remark:** Combining the above result and that in Theorem 4.2, Algorithm 2 needs $T = \mathcal{O}(\frac{1}{\epsilon^6})$ to achieve a nearly $\epsilon$-KKT solution to the original problem equation 1. It is better than the complexity of $\mathcal{O}(1/\epsilon^7)$ of the single-loop algorithm considered in Li et al. (2024a).

**Theorem D.3.** *Suppose Assumption 3.1, 5.1 and D.1 with condition (ii) hold. Let* $\gamma = \mathcal{O}(\min\{\frac{|\mathcal{B}_{2,k}|}{\beta^4}\epsilon^4, \frac{|\mathcal{B}_{1,i}|}{\beta}\epsilon^2\})$, $\eta = \mathcal{O}(\min\{\frac{|\mathcal{B}||\mathcal{B}_{1,i}|^{1/2}\epsilon^2}{n\beta^2}, \frac{|\mathcal{B}_c||\mathcal{B}_{2,k}|^{1/2}\epsilon^4}{\beta^5 m}\})$. *After* $T = \mathcal{O}(\max\{\frac{m\beta^6}{|\mathcal{B}_{2,k}|^{1/2}|\mathcal{B}_c|\epsilon^6}, \frac{n\beta^3}{|\mathcal{B}||\mathcal{B}_{1,i}|^{1/2}\epsilon^4}, \frac{n\beta^2}{|\mathcal{B}||\mathcal{B}_{1,i}|\epsilon^4}\})$ *iterations, the Algorithm 2 satisfies* $\mathbb{E}[\|\nabla\Phi_\lambda(\mathbf{x}_{\hat{t}})\|] \leq \epsilon$ *from some* $\lambda = O(1/(\rho_0 + \rho_1\beta))$, *where* $\hat{t}$ *is selected uniformly at random from* $\{1, \ldots, T\}$

*Proof of Theorem D.2. .*

The objective function is $\min_{\mathbf{x}} \Phi(\mathbf{x}, \mathcal{D}) := \underbrace{\frac{1}{n}\sum_{i=1}^n f_i(g_i(\mathbf{x}))}_{F(\mathbf{x})} + \underbrace{\frac{\beta}{m}\sum_{k=1}^m [h_k(\mathbf{x})]_+}_{H(\mathbf{x})}$.

For simplicity, denote $\bar{\mathbf{x}}_t := \mathrm{prox}_{\Phi/\bar{\rho}}(\mathbf{x}_t)$. Consider change in the Moreau envelope

$$\mathbb{E}_t[\Phi_{1/\bar{\rho}}(\mathbf{x}_{t+1})] = \mathbb{E}_t[\min_{\tilde{\mathbf{x}}} \Phi(\tilde{\mathbf{x}}) + \frac{\bar{\rho}}{2}\|\tilde{\mathbf{x}} - \mathbf{x}_{t+1}\|^2]$$

$$\leq \mathbb{E}_t[\Phi(\bar{\mathbf{x}}_t) + \frac{\bar{\rho}}{2}\|\bar{\mathbf{x}}_t - \mathbf{x}_{t+1}\|^2]$$

$$= \mathbb{E}_t[\Phi(\bar{\mathbf{x}}_t) + \frac{\bar{\rho}}{2}\|\bar{\mathbf{x}}_t - (\mathbf{x}_t - \eta(G_1^t + G_2^t))\|^2]$$

$$\leq \Phi(\bar{\mathbf{x}}_t) + \frac{\bar{\rho}}{2}\|\bar{\mathbf{x}}_t - \mathbf{x}_t\|^2 + \bar{\rho}\mathbb{E}_t[\eta\langle\bar{\mathbf{x}}_t - \mathbf{x}_t, G_1^t + G_2^t\rangle] + \eta^2\frac{\bar{\rho}M^2}{2}$$

$$= \Phi_{1/\bar{\rho}}(\mathbf{x}_t) + \bar{\rho}\eta\langle\bar{\mathbf{x}}_t - \mathbf{x}_t, \mathbb{E}_t[G_1^t]\rangle + \bar{\rho}\eta\langle\bar{\mathbf{x}}_t - \mathbf{x}_t, \mathbb{E}_t[G_2^t]\rangle + \eta^2\frac{\bar{\rho}M^2}{2}, \quad (28)$$

where the second inequality uses the bound of $\mathbb{E}_t[\|G_1^t + G_2^t\|^2]$, which follows from the Lipchitz continuity, denoted by $M^2 \geq 2L_f^2 L_g^2 + 2\beta^2 L_h^2$.

$$\mathbb{E}_t[G_1^t] \in \frac{1}{n}\sum_{i=1}^n \partial f_i(u_{1,i}^t)^\top \partial g_i(\mathbf{x}_t)^\top, \quad \mathbb{E}_t[G_2^t] \in \frac{\beta}{m}\sum_{k=1}^m \partial h_k(\mathbf{x}_t)[u_{2,k}^t]_+'.$$

Next we give the estimation of $\langle\bar{\mathbf{x}}_t - \mathbf{x}_t, \mathbb{E}_t[G_1^t]\rangle$ and $\langle\bar{\mathbf{x}}_t - \mathbf{x}_t, \mathbb{E}_t[G_2^t]\rangle$. For given $i \in \{1, \ldots, n\}$, we have

$$f_i(g_i(\bar{\mathbf{x}}_t)) - f_i(u_{1,i}^t) \geq \partial f_i(u_{1,i}^t)^\top(g_i(\bar{\mathbf{x}}_t) - u_{1,i}^t) - \frac{\rho_f}{2}\|g_i(\bar{\mathbf{x}}_t) - u_{1,i}^t\|^2$$

$$\geq \partial f_i(u_{1,i}^t)^\top(g_i(\bar{\mathbf{x}}_t) - u_{1,i}^t) - \rho_f\|g_i(\bar{\mathbf{x}}_t) - g_i(\mathbf{x}_t)\|^2 - \rho_f\|g_i(\mathbf{x}_t) - u_{1,i}^t\|^2$$

$$\geq \partial f_i(u_{1,i}^t)^\top \left[g_i(\mathbf{x}_t) - u_{1,i}^t + \partial g_i(\mathbf{x}_t)^\top(\bar{\mathbf{x}}_t - \mathbf{x}_t) - \frac{\rho_g}{2}\|\bar{\mathbf{x}}_t - \mathbf{x}_t\|^2\right] - \rho_f L_g^2\|\bar{\mathbf{x}}_t - \mathbf{x}_t\|^2 - \rho_f\|g_i(\mathbf{x}_t) - u_{1,i}^t\|^2$$

$$\geq \partial f_i(u_{1,i}^t)^\top(g_i(\mathbf{x}_t) - u_{1,i}^t) + \partial f_i(u_{1,i}^t)^\top \partial g_i(\mathbf{x}_t)^\top(\bar{\mathbf{x}}_t - \mathbf{x}_t) - (\frac{\rho_g L_f}{2} + \rho_f L_g^2)\|\bar{\mathbf{x}}_t - \mathbf{x}_t\|^2 - \rho_f\|g_i(\mathbf{x}_t) - u_{1,i}^t\|^2,$$

where the first inequality holds by $\rho_f$-weakly convex of $f_i$, the second inequality holds by the monotonically non-decreasing of $f_i$ and the third inequality holds by the $\rho_g$-weakly convexity of $g_i$, the last inequality holds by and $0 \leq \partial f_i(u_i^t) \leq L_f$. Then it follows

$$\frac{1}{n}\sum_{i=1}^n \partial f_i(u_{1,i}^t)^\top \partial g_i(\mathbf{x}_t)^\top(\bar{\mathbf{x}}_t - \mathbf{x}_t) \leq \frac{1}{n}\sum_{i=1}^n \left[f_i(g_i(\bar{\mathbf{x}}_t)) - f_i(u_{1,i}^t) - \partial f_i(u_{1,i}^t)^\top(g_i(\mathbf{x}_t) - u_{1,i}^t)\right.$$

$$\left. + (\frac{\rho_g L_f}{2} + \rho_f L_g^2)\|\bar{\mathbf{x}}_t - \mathbf{x}_t\|^2 + \rho_f\|g_i(\mathbf{x}_t) - u_{1,i}^t\|^2\right]. \quad (29)$$

Next we bound $\langle\bar{\mathbf{x}}_t - \mathbf{x}_t, \mathbb{E}_t[G_2^t]\rangle$. For given $k \in \{1, \ldots, m\}$, from equation 20, we have

$$\beta\frac{1}{m}\sum_{k=1}^m [u_{2,k}^t]_+' \partial h_k(\mathbf{x}_t)^\top(\bar{\mathbf{x}}_t - \mathbf{x}_t) \leq \frac{\beta}{m}\sum_{k=1}^m \left[[h_k(\bar{\mathbf{x}}_t)]_+ - [u_{2,k}^t]_+ - [u_{2,k}^t]_+'(h_k(\mathbf{x}_t) - u_{2,k}^t) + \frac{\rho_1}{2}\|\bar{\mathbf{x}}_t - \mathbf{x}_t\|^2\right].$$

$$(30)$$

Combining inequality equation 28, equation 29 and equation 30 yields

$$\mathbb{E}_t[\Phi_{1/\bar{\rho}}(\mathbf{x}_{t+1})] \leq \Phi_{1/\bar{\rho}}(\mathbf{x}_t) + \bar{\rho}\eta\langle\bar{\mathbf{x}}_t - \mathbf{x}_t, \mathbb{E}_t[G_1^t]\rangle + \bar{\rho}\eta\langle\bar{\mathbf{x}}_t - \mathbf{x}_t, \mathbb{E}_t[G_2^t]\rangle + \eta^2\frac{\bar{\rho}M^2}{2}$$

$$\leq \Phi_{1/\bar{\rho}}(\mathbf{x}_t) + \eta^2\frac{\bar{\rho}M^2}{2}$$

$$+ \bar{\rho}\eta\frac{1}{n}\sum_{i=1}^{n}\left[f_i(g_i(\bar{\mathbf{x}}_t)) - f_i(u_{1,i}^t) - \partial f_i(u_{1,i}^t)^\top(g_i(\mathbf{x}_t) - u_{1,i}^t) + (\frac{\rho_g L_f}{2} + \rho_f L_g^2)\|\bar{\mathbf{x}}_t - \mathbf{x}_t\|^2 + \rho_f\|g_i(\mathbf{x}_t) - u_{1,i}^t\|^2\right]$$

$$+ \bar{\rho}\eta\frac{\beta}{m}\sum_{k=1}^{m}\left[[h_k(\bar{\mathbf{x}}_t)]_+ - [u_{2,k}^t]_+ - [u_{2,k}^t]_+'(h_k(\mathbf{x}_t) - u_{2,k}^t) + \frac{\rho_1}{2}\|\bar{\mathbf{x}}_t - \mathbf{x}_t\|^2\right]$$

$$= \Phi_{1/\bar{\rho}}(\mathbf{x}_t) + \eta^2\frac{\bar{\rho}M^2}{2} + \bar{\rho}\eta\frac{1}{n}\sum_{i=1}^{n}(\frac{\rho_g L_f}{2} + \rho_f L_g^2)\|\bar{\mathbf{x}}_t - \mathbf{x}_t\|^2 + \bar{\rho}\eta\frac{\beta}{m}\sum_{k=1}^{m}\frac{\rho_1}{2}\|\bar{\mathbf{x}}_t - \mathbf{x}_t\|^2$$

$$+ \bar{\rho}\eta\frac{1}{n}\sum_{i=1}^{n}\left[f_i(g_i(\bar{\mathbf{x}}_t)) - f_i(g_i(\mathbf{x}_t)) + f_i(g_i(\mathbf{x}_t)) - f_i(u_{1,i}^t) - \partial f_i(u_{1,i}^t)^\top(g_i(\mathbf{x}_t) - u_i^t) + \rho_f\|g_i(\mathbf{x}_t) - u_{1,i}^t\|^2\right]$$

$$+ \bar{\rho}\eta\frac{\beta}{m}\sum_{k=1}^{m}\left[[h_k(\bar{\mathbf{x}}_t)]_+ - [h_k(\mathbf{x}_t)]_+ + [h_k(\mathbf{x}_t)]_+ - [u_{2,k}^t]_+ - [u_{2,k}^t]_+'(h_k(\mathbf{x}_t) - u_{2,k}^t)\right]. \tag{31}$$

From Lemma 4.1, the function $\Phi$ is $C$-weakly convex. We have $(\bar{\rho} - C)-$strong convexity of $\mathbf{x} \mapsto \Phi(\mathbf{x}) + \frac{\bar{\rho}}{2}\|\mathbf{x}_t - \mathbf{x}\|^2$

$$\Phi(\bar{\mathbf{x}}_t) - \Phi(\mathbf{x}_t) \leq (\frac{C}{2} - \bar{\rho})\|\bar{\mathbf{x}}_t - \mathbf{x}_t\|^2.$$

We get

$$\frac{1}{n}\sum_{i=1}^{n}f_i(g_i(\bar{\mathbf{x}}_t)) - f_i(g_i(\mathbf{x}_t)) + \beta\frac{1}{m}\sum_{k=1}^{m}([h_k(\bar{\mathbf{x}}_t)]_+ - [h_k(\mathbf{x}_t)]_+) = \Phi(\bar{\mathbf{x}}_t) - \Phi(\mathbf{x}_t) \leq (\frac{C}{2} - \bar{\rho})\|\bar{\mathbf{x}}_t - \mathbf{x}_t\|^2. \tag{32}$$

Plugging inequalities equation 32 to equation 31, we get

$$\mathbb{E}_t[\Phi_{1/\bar{\rho}}(\mathbf{x}_{t+1})]$$

$$\leq \Phi_{1/\bar{\rho}}(\mathbf{x}_t) + \eta^2\frac{\bar{\rho}M^2}{2} + \bar{\rho}\eta\left(\frac{\rho_g L_f}{2} + \rho_f L_g^2 + \beta\frac{\rho_1}{2} + \frac{C}{2} - \bar{\rho}\right)\|\bar{\mathbf{x}}_t - \mathbf{x}_t\|^2$$

$$+ \bar{\rho}\eta\frac{1}{n}\sum_{i=1}^{n}\left[f_i(g_i(\mathbf{x}_t)) - f_i(u_{1,i}^t)) - \partial f_i(u_{1,i}^t)^\top(g_i(\mathbf{x}_t) - u_{1,i}^t) + \rho_f\|g_i(\mathbf{x}_t) - u_{1,i}^t\|^2\right]$$

$$+ \bar{\rho}\eta\frac{\beta}{m}\sum_{k=1}^{m}\left[[h_k(\mathbf{x}_t)]_+ - [u_{2,k}^t]_+ - [u_{2,k}^t]_+'(h_k(\mathbf{x}_t) - u_{2,k}^t)\right].$$

Set $\frac{\rho_g L_f}{2} + \rho_f L_g^2 + \beta \frac{\rho_1}{2} + \frac{C}{2} = \frac{\bar{\rho}}{2}$, we have

$$\mathbb{E}_t[\Phi_{1/\bar{\rho}}(\mathbf{x}_{t+1})]$$

$$\leq \Phi_{1/\bar{\rho}}(\mathbf{x}_t) + \eta^2 \frac{\bar{\rho} M^2}{2} - \eta \frac{\bar{\rho}^2}{2} \|\bar{\mathbf{x}}_t - \mathbf{x}_t\|^2$$

$$+ \bar{\rho}\eta \frac{1}{n} \sum_{i=1}^{n} \left[ f_i(g_i(\mathbf{x}_t)) - f_i(u_{1,i}^t) - \partial f_i(u_{1,i}^t)^\top (g_i(\mathbf{x}_t) - u_{1,i}^t) + \rho_f \|g_i(\mathbf{x}_t) - u_{1,i}^t\|^2 \right]$$

$$+ \bar{\rho}\eta \frac{\beta}{m} \sum_{k=1}^{m} \left[ [h_k(\mathbf{x}_t)]_+ - [u_{2,k}^t]_+ - [u_{2,k}^t]'_+ (h_k(\mathbf{x}_t) - u_{2,k}^t) \right]$$

$$\leq \Phi_{1/\bar{\rho}}(\mathbf{x}_t) + \eta^2 \frac{\bar{\rho} M^2}{2} - \frac{\eta}{2} \|\nabla \Phi_{1/\bar{\rho}}(\mathbf{x}_t)\|^2$$

$$+ \bar{\rho}\eta \frac{1}{n} \sum_{i=1}^{n} \left[ f_i(g_i(\mathbf{x}_t)) - f_i(u_{1,i}^t) - \partial f_i(u_{1,i}^t)^\top (g_i(\mathbf{x}_t) - u_{1,i}^t) + \rho_f \|g_i(\mathbf{x}_t) - u_{1,i}^t\|^2 \right]$$

$$+ \bar{\rho}\eta \frac{\beta}{m} \sum_{k=1}^{m} \left[ [h_k(\mathbf{x}_t)]_+ - [u_{2,k}^t]_+ - [u_{2,k}^t]'_+ (h_k(\mathbf{x}_t) - u_{2,k}^t) \right].$$

Using the Lipschitz continuity of $f_i$ and the fact that $0 \leq [u_{2,k}^t]'_+ \leq 1$, we get

$$\mathbb{E}_t[\Phi_{1/\bar{\rho}}(\mathbf{x}_{t+1})] \leq \Phi_{1/\bar{\rho}}(\mathbf{x}_t) + \eta^2 \frac{\bar{\rho} M^2}{2} - \frac{\eta}{2} \|\nabla \Phi_{1/\bar{\rho}}(\mathbf{x}_t)\|^2 + 2\bar{\rho} L_f \eta \frac{1}{n} \sum_{i=1}^{n} \|g_i(\mathbf{x}_t) - u_{1,i}^t\|$$

$$+ \bar{\rho}\rho_f \eta \frac{1}{n} \sum_{i=1}^{n} \|g_i(\mathbf{x}_t) - u_{1,i}^t\|^2 + 2\bar{\rho}\eta \frac{\beta}{m} \sum_{k=1}^{m} \|h_k(\mathbf{x}_t) - u_{2,k}^t\|. \tag{33}$$

Taking the full expectation on both sides of equation 33 and adding the result of Lemma B.2, equation 14 and equation 13, we have

$$\mathbb{E}[\Phi_{1/\bar{\rho}}(\mathbf{x}_{t+1})]$$

$$\leq \mathbb{E}[\Phi_{1/\bar{\rho}}(\mathbf{x}_t)] + \eta^2 \frac{\bar{\rho} M^2}{2} - \frac{\eta}{2} \mathbb{E}[\|\nabla \Phi_{1/\bar{\rho}}(\mathbf{x}_t)\|^2]$$

$$+ 2\bar{\rho}\eta \left[ L_f (1 - \frac{|\mathcal{B}|\gamma_1}{2n})^t \frac{1}{n} \sum_{i=1}^{n} \|u_{1,i}^0 - g_i(\mathbf{x}_0)\| + \frac{4n L_g L_f M \eta}{|\mathcal{B}|\gamma_1^{1/2}} + \frac{2 L_f \gamma_1^{1/2} \sigma_g}{|\mathcal{B}_{1,i}|^{1/2}} \right]$$

$$+ \bar{\rho}\eta \left[ \rho_f (1 - \frac{|\mathcal{B}|\gamma_1}{n})^t \frac{1}{n} \sum_{i=1}^{n} \|u_{1,i}^0 - g_i(\mathbf{x}_0)\|^2 + \frac{8n^2\eta^2 L_g^2 \rho_f M^2}{|\mathcal{B}|^2 \gamma_1} + \frac{2\gamma_1 \sigma_g^2 \rho_f}{|\mathcal{B}_{1,i}|} \right]$$

$$+ 2\bar{\rho}\eta\beta \left[ (1 - \frac{|\mathcal{B}_c|\gamma_2}{2m})^t \frac{1}{m} \sum_{k=1}^{m} \|u_{2,k}^0 - h_k(\mathbf{x}_0)\| + \frac{4m L_h \eta M}{|\mathcal{B}_c|\gamma_2^{1/2}} + \frac{2\gamma_2^{1/2} \sigma_h}{|\mathcal{B}_{2,k}|^{1/2}} \right].$$

Taking summation from $t = 0$ to $T - 1$ yields

$$\mathbb{E}[\Phi_{1/\bar{\rho}}(\mathbf{x}_T)]$$

$$\leq \Phi_{1/\bar{\rho}}(\mathbf{x}_0) + \eta^2 T \frac{\bar{\rho} M^2}{2} - \frac{\eta}{2} \sum_{t=0}^{T-1} \mathbb{E}[\|\nabla \Phi_{1/\bar{\rho}}(\mathbf{x}_t)\|^2]$$

$$+ 2\bar{\rho}\eta \left[ L_f \frac{2n}{|\mathcal{B}|\gamma_1} \frac{1}{n} \sum_{i=1}^{n} \|u_{1,i}^0 - g_i(\mathbf{x}_0)\| + T \frac{8n L_g L_f \eta M}{|\mathcal{B}|\gamma_1^{1/2}} + T \frac{4 L_f \gamma_1^{1/2} \sigma_g}{|\mathcal{B}_{1,i}|^{1/2}} \right]$$

$$+ \bar{\rho}\eta \left[ \rho_f \frac{n}{|\mathcal{B}|\gamma_1} \frac{1}{n} \sum_{i=1}^{n} \|u_{1,i}^0 - g_i(\mathbf{x}_0)\|^2 + T \frac{8n^2\eta^2 L_g^2 \rho_f M^2}{|\mathcal{B}|^2 \gamma_1} + T \frac{2\gamma_1 \sigma_g^2 \rho_f}{|\mathcal{B}_{1,i}|} \right]$$

$$+ 2\bar{\rho}\eta\beta \left[ \frac{2m}{|\mathcal{B}_c|\gamma_2} \frac{1}{m} \sum_{k=1}^{m} \|u_{2,k}^0 - h_k(\mathbf{x}_0)\| + T \frac{8m L_h \eta M}{|\mathcal{B}_c|\gamma_2^{1/2}} + T \frac{4\gamma_2^{1/2} \sigma_h}{|\mathcal{B}_{2,k}|^{1/2}} \right],$$

where we use the fact that $\sum_{t=0}^{T-1}(1-\mu) \leq \frac{1}{\mu}$. Lower bounding the left-hand-side by $\min_{\mathbf{x}} \Phi(\mathbf{x})$ and dividing both sides by $T$, we obtain

$$
\frac{1}{T} \sum_{t=0}^{T-1} \mathbb{E}[\|\nabla \Phi_{1/\bar{\rho}}(\mathbf{w}_t)\|^2]
$$

$$
\leq \frac{2}{\eta T}[\Phi_{1/\bar{\rho}}(\mathbf{x}_0) - \min_{\mathbf{x}} \Phi(\mathbf{x})] + \eta \bar{\rho} M^2 + \frac{\mathcal{C}\bar{\rho}}{T\gamma} \max\{\frac{n}{|\mathcal{B}|}, \frac{\beta m}{|\mathcal{B}_c|}\}
$$

$$
+ \mathcal{C}\Big(\bar{\rho}\frac{\eta M}{\gamma^{1/2}} \max\{\frac{n}{|\mathcal{B}|}, \frac{\beta m}{|\mathcal{B}_c|}\} + \frac{n^2 \eta^2 M^2 \bar{\rho}}{|\mathcal{B}|^2 \gamma} + \gamma^{1/2} \bar{\rho} \max\{\frac{1}{|\mathcal{B}_{1,i}|^{1/2}}, \frac{\beta}{|\mathcal{B}_{2,k}|^{1/2}}\} + \bar{\rho}\frac{\gamma}{|\mathcal{B}_{1,i}|}\Big).
$$

where we set $\gamma_1 = \gamma_2 = \gamma$ and

$$
\mathcal{C} = \max\{8L_f \frac{1}{n} \sum_{i=1}^{n} \|u_{1,i}^0 - g_i(\mathbf{x}_0)\|, 2\rho_f \frac{1}{n} \sum_{i=1}^{n} \|u_{1,i}^0 - g_i(\mathbf{x}_0)\|^2, 8\frac{1}{m} \sum_{k=1}^{m} \|u_{2,k}^0 - h_k(\mathbf{x}_0)\|
$$

$$
32L_g L_f, 16L_g^2 \rho_f, 16L_f \sigma_g, 4\sigma_g^2 \rho_f, 16L_h, 16\sigma_h\}.
$$

As we set $M^2 \geq 2L_f^2 L_g^2 + 2\beta^2 L_h^2$ and $\frac{\rho_g L_f}{2} + \rho_f L_g^2 + \beta \frac{\rho_1}{2} + \frac{C}{2} = \frac{\bar{\rho}}{2}$, with $\gamma = \mathcal{O}(\min\{|\mathcal{B}_{1,i}|, \frac{|\mathcal{B}_{2,k}|}{\beta^2}\}\frac{\epsilon^4}{\beta^2})$, $\eta = \mathcal{O}(\min\{\frac{|\mathcal{B}|}{n}, \frac{|\mathcal{B}_c|}{\beta m}\} \min\{|\mathcal{B}_{1,i}|^{1/2}, \frac{|\mathcal{B}_{2,k}|^{1/2}}{\beta}\}\frac{\epsilon^4}{\beta^3})$ and $T = \mathcal{O}(\max\{\frac{\beta}{|\mathcal{B}_{1,i}|^{1/2}}, \frac{\beta^2}{|\mathcal{B}_{2,k}|^{1/2}}, \frac{1}{|\mathcal{B}_{1,i}|}\} \max\{\frac{n}{|\mathcal{B}|}, \frac{\beta m}{|\mathcal{B}_c|}\}\frac{\beta^3}{\epsilon^6})$, we have

$$
\frac{1}{T} \sum_{t=0}^{T-1} \mathbb{E}[\|\nabla \Phi_{1/\bar{\rho}}(\mathbf{x}_t)\|^2] \leq \mathcal{O}(\epsilon^2).
$$

By Jensen's inequality, we can get $\mathbb{E}[\|\nabla \Phi_{1/\bar{\rho}}(\mathbf{x}_{\hat{t}})\|] \leq \mathcal{O}(\epsilon)$, where $\hat{t}$ is selected uniformly at random from $\{1, \ldots, T\}$. $\qquad\square$

Now, we give the proof of Theorem D.3.

*Proof of Theorem D.3.* From equation 28, also denote $\bar{\mathbf{x}}_t := \text{prox}_{\Phi/\bar{\rho}}(\mathbf{x}_t)$ and consider change in the Moreau envelope, we have

$$
\mathbb{E}_t[\Phi_{1/\bar{\rho}}(\mathbf{x}_{t+1})] \leq \Phi_{1/\bar{\rho}}(\mathbf{x}_t) + \bar{\rho}\eta\langle\bar{\mathbf{x}}_t - \mathbf{x}_t, \mathbb{E}_t[G_1^t]\rangle + \bar{\rho}\eta\langle\bar{\mathbf{x}}_t - \mathbf{x}_t, \mathbb{E}_t[G_2^t]\rangle + \eta^2\frac{\bar{\rho}M^2}{2}, \tag{34}
$$

where the gradient can be bounded by $M$, where $M^2 \geq 2L_f^2 L_g^2 + 2\beta^2 L_h^2$ and

$$
\mathbb{E}_t[G_1^t] \in \frac{1}{n} \sum_{i=1}^{n} \nabla f_i(u_{1,i}^t)^\top \nabla g_i(\mathbf{x}_t)^\top, \qquad \mathbb{E}_t[G_2^t] \in \frac{\beta}{m} \sum_{k=1}^{m} \partial h_k(\mathbf{x}_t)[u_{2,k}^t]_+'.
$$

Next we give the bound of $\langle\bar{\mathbf{x}}_t - \mathbf{x}_t, \mathbb{E}_t[G_2^t]\rangle$ and $\langle\bar{\mathbf{x}}_t - \mathbf{x}_t, \mathbb{E}_t[G_2^t]\rangle$. It's easy to verify that $F$ is $L_{\nabla F}$-smooth, where $L_{\nabla F} = L_f L_{\nabla g} + L_g^2 L_{\nabla f}$. Then we have

$$
F(\mathbf{x}_t) \leq F(\bar{\mathbf{x}}_t) + \langle\nabla F(\bar{\mathbf{x}}_t), \mathbf{x}^t - \bar{\mathbf{x}}_t\rangle + \frac{L_{\nabla F}}{2}\|\bar{\mathbf{x}}_t - \mathbf{x}_t\|^2
$$

$$
= F(\bar{\mathbf{x}}_t) + \langle\nabla F(\bar{\mathbf{x}}_t) - \nabla F(\mathbf{x}_t), \mathbf{x}_t - \bar{\mathbf{x}}_t\rangle + \frac{L_{\nabla F}}{2}\|\bar{\mathbf{x}}_t - \mathbf{x}_t\|^2
$$

$$
+ \langle\frac{1}{n} \sum_{i=1}^{n} \nabla f_i(g_i(\mathbf{x}_t))^\top \nabla g_i(\mathbf{x}_t)^\top - \frac{1}{n} \sum_{i=1}^{n} \nabla f_i(u_{1,i}^t)^\top \nabla g_i(\mathbf{x}_t)^\top, \mathbf{x}_t - \bar{\mathbf{x}}_t\rangle
$$

$$
+ \langle\frac{1}{n} \sum_{i=1}^{n} \nabla f_i(u_{1,i}^t)^\top \nabla g_i(\mathbf{x}_t)^\top, \mathbf{x}_t - \bar{\mathbf{x}}_t\rangle.
$$

Then, we have

$$\mathbb{E}_t[\langle G_1^t, \bar{\mathbf{x}}_t - \mathbf{x}_t \rangle] = \langle \frac{1}{n} \sum_{i=1}^n \nabla f_i(u_{1,i}^t)^\top \nabla g_i(\mathbf{x}_t)^\top, \bar{\mathbf{x}}_t - \mathbf{x}_t \rangle$$

$$\leq F(\bar{\mathbf{x}}_t) - F(\mathbf{x}_t) + \frac{L_{\nabla F}}{2} \|\bar{\mathbf{x}}_t - \mathbf{x}_t\|^2 + L_{\nabla F} \|\bar{\mathbf{x}}_t - \mathbf{x}_t\|^2 + L_{\nabla f} L_g \frac{1}{n} \sum_{i=1}^n \|g_i(\mathbf{x}_t) - u_{1,i}^t\| \|\bar{\mathbf{x}}_t - \mathbf{x}_t\|$$

$$\leq F(\bar{\mathbf{x}}_t) - F(\mathbf{x}_t) + (2L_{\nabla F} + 2L_{\nabla f} L_g) \|\bar{\mathbf{x}}_t - \mathbf{x}_t\|^2 + 2L_{\nabla f} L_g \frac{1}{n} \sum_{i=1}^n \|g_i(\mathbf{x}_t) - u_{1,i}^t\|^2.$$

From the equation 20, we have

$$\mathbb{E}_t[\langle G_2^t, \bar{\mathbf{x}}_t - \mathbf{x}_t \rangle] = \beta \frac{1}{m} \sum_{k=1}^m [u_{2,k}^t]_+' \partial h_k(\mathbf{x}_t)^\top (\bar{\mathbf{x}}_t - \mathbf{x}_t)$$

$$\leq \frac{\beta}{m} \sum_{k=1}^m \left[ [h_k(\bar{\mathbf{x}}_t)]_+ - [u_{2,k}^t]_+ - [u_{2,k}^t]_+'(h_k(\mathbf{x}_t) - u_{2,k}^t) + \frac{\rho_1}{2} \|\bar{\mathbf{x}}_t - \mathbf{x}_t\|^2 \right].$$

Adding above two estimation to equation 34, we have

$$\mathbb{E}_t[\Phi_{1/\bar{\rho}}(\mathbf{x}_{t+1})] \leq \Phi_{1/\bar{\rho}}(\mathbf{x}_t) + \bar{\rho}\eta \langle \bar{\mathbf{x}}_t - \mathbf{x}_t, \mathbb{E}_t[G_1^t] \rangle + \bar{\rho}\eta \langle \bar{\mathbf{x}}_t - \mathbf{x}_t, \mathbb{E}_t[G_2^t] \rangle + \eta^2 \frac{\bar{\rho}M^2}{2}$$

$$\leq \Phi_{1/\bar{\rho}}(\mathbf{x}_t) + \eta^2 \frac{\bar{\rho}M^2}{2}$$

$$+ \bar{\rho}\eta \left[ F(\bar{\mathbf{x}}_t) - F(\mathbf{x}_t) + (2L_{\nabla F} + 2L_{\nabla f} L_g) \|\bar{\mathbf{x}}_t - \mathbf{x}_t\|^2 + 2L_{\nabla f} L_g \frac{1}{n} \sum_{i=1}^n \|g_i(\mathbf{x}_t) - u_{1,i}^t\|^2 \right]$$

$$+ \bar{\rho}\eta \frac{\beta}{m} \sum_{k=1}^m \left[ [h_k(\bar{\mathbf{x}}_t)]_+ - [h_k(\mathbf{x}_t)]_+ + [h_k(\mathbf{x}_t)]_+ - [u_{2,k}^t]_+ - [u_{2,k}^t]_+'(h_k(\mathbf{x}_t) - u_{2,k}^t) + \frac{\rho_1}{2} \|\bar{\mathbf{x}}_t - \mathbf{x}_t\|^2 \right]. \quad (35)$$

Since smoothness of $F$ implies the weakly convexity of $F$, we still have $(\bar{\rho} - C)-$strong convexity of $\mathbf{x} \mapsto \Phi(\mathbf{x}) + \frac{\bar{\rho}}{2} \|\mathbf{x}_t - \mathbf{x}\|^2$

$$\Phi(\bar{\mathbf{x}}_t) - \Phi(\mathbf{x}_t) \leq (\frac{C}{2} - \bar{\rho}) \|\bar{\mathbf{x}}_t - \mathbf{x}_t\|^2.$$

Adding above inequality to equation 35, we have

$$\mathbb{E}_t[\Phi_{1/\bar{\rho}}(\mathbf{x}_{t+1})] \leq \Phi_{1/\bar{\rho}}(\mathbf{x}_t) + \eta^2 \frac{\bar{\rho}M^2}{2} + \bar{\rho}\eta(2L_{\nabla F} + 2L_{\nabla f} L_g + \beta \frac{\rho_1}{2} + \frac{C}{2} - \bar{\rho}) \|\bar{\mathbf{x}}_t - \mathbf{x}_t\|^2$$

$$+ 2L_{\nabla f} L_g \bar{\rho}\eta \frac{1}{n} \sum_{i=1}^n \|g_i(\mathbf{x}_t) - u_{1i}^t\|^2 + \bar{\rho}\eta \frac{\beta}{m} \sum_{k=1}^m \left[ [h_k(\mathbf{x}_t)]_+ - [u_{2,k}^t]_+ - [u_{2,k}^t]_+'(h_k(\mathbf{x}_t) - u_{2,k}^t) \right].$$

If we set $2L_{\nabla F} + 2L_{\nabla f} L_f + \beta \frac{\rho_1}{2} + \frac{C}{2} = \frac{\bar{\rho}}{2}$, we have

$$\mathbb{E}_t[\Phi_{1/\bar{\rho}}(\mathbf{x}_{t+1})] \leq \Phi_{1/\bar{\rho}}(\mathbf{x}_t) + \eta^2 \frac{\bar{\rho}M^2}{2} - \bar{\rho}\eta \frac{\bar{\rho}}{2} \|\bar{\mathbf{x}}_t - \mathbf{x}_t\|^2$$

$$+ 2L_{\nabla f} L_g \bar{\rho}\eta \frac{1}{n} \sum_{i=1}^n \|g_i(\mathbf{x}_t) - u_{1i}^t\|^2 + \bar{\rho}\eta \frac{\beta}{m} \sum_{k=1}^m \left[ [h_k(\mathbf{x}_t)]_+ - [u_{2,k}^t]_+ - [u_{2,k}^t]_+'(h_k(\mathbf{x}_t) - u_{2,k}^t) \right]$$

$$\leq \Phi_{1/\bar{\rho}}(\mathbf{x}_t) + \eta^2 \frac{\bar{\rho}M^2}{2} - \frac{\eta}{2} \|\nabla \Phi_{1/\bar{\rho}}(\mathbf{x}_t)\|^2 + 2L_{\nabla f} \bar{\rho}\eta \frac{1}{n} \sum_{i=1}^n \|g_i(\mathbf{x}_t) - u_{1i}^t\|^2 + 2\bar{\rho}\eta \frac{\beta}{m} \sum_{k=1}^m \|h_k(\mathbf{x}_t) - u_{2,k}^t\|,$$

where we use the fact $0 \leq [u_{2,k}^t]_+' \leq 1$.

Adding the result of Lemma B.2 and inequality equation 13, we get

$$
\mathbb{E}[\Phi_{1/\bar{\rho}}(\mathbf{x}_{t+1})] \leq \mathbb{E}[\Phi_{1/\bar{\rho}}(\bar{\mathbf{x}}_t)] + \eta^2 \frac{\bar{\rho}M^2}{2} - \frac{\eta}{2}\mathbb{E}[\|\nabla\Phi_{1/\bar{\rho}}(\mathbf{x}_t)\|^2]
$$
$$
+ 2\bar{\rho}\eta\Big[L_{\nabla f}L_g(1-\frac{|\mathcal{B}|\gamma_1}{n})^t\frac{1}{n}\sum_{i=1}^n\|u_{1,i}^0 - g_i(\mathbf{x}_0)\|^2 + \frac{8n^2\eta^2 L_g^2 L_{\nabla f}L_g M^2}{|\mathcal{B}|^2\gamma_1} + \frac{2\gamma_1\sigma_g^2 L_{\nabla f}L_g}{|\mathcal{B}_{1,i}|}\Big]
$$
$$
+ 2\bar{\rho}\eta\beta\Big[(1-\frac{|\mathcal{B}_c|\gamma_2}{2m})^t\frac{1}{m}\sum_{k=1}^m\|u_{2,k}^0 - h_k(\mathbf{x}_0)\| + \frac{4mL_h^2\eta M}{|\mathcal{B}_c|\gamma_2^{1/2}} + \frac{2\gamma_2^{1/2}\sigma_h}{|\mathcal{B}_{2,k}|^{1/2}}\Big].
$$

Taking summation from $t=0$ to $T-1$ yields

$$
\mathbb{E}[\Phi_{1/\bar{\rho}}(\mathbf{x}_T)]
$$
$$
\leq \mathbb{E}[\Phi_{1/\bar{\rho}}(\mathbf{x}_0)] + \eta^2 T\frac{\bar{\rho}M^2}{2} - \frac{\eta}{2}\sum_{t=0}^{T-1}\mathbb{E}[\|\nabla\Phi_{1/\bar{\rho}}(\mathbf{x}_t)\|^2]
$$
$$
+ 2\bar{\rho}\eta\Big[L_{\nabla f}L_g\frac{n}{|\mathcal{B}|\gamma_1}\frac{1}{n}\sum_{i=1}^n\|u_{1,i}^0 - g_i(\mathbf{x}_0)\|^2 + T\frac{8n^2\eta^2 L_g^3 L_{\nabla f}M^2}{|\mathcal{B}|^2\gamma_1} + T\frac{2\gamma_1\sigma_g^2 L_{\nabla f}L_g}{|\mathcal{B}_{1,i}|}\Big]
$$
$$
+ 2\bar{\rho}\eta\beta\Big[\frac{2m}{|\mathcal{B}_c|\gamma_2}\frac{1}{m}\sum_{k=1}^m\|u_{2,k}^0 - h_k(\mathbf{x}_0)\| + T\frac{8mL_h\eta M}{|\mathcal{B}_c|\gamma_2^{1/2}} + T\frac{4\gamma_2^{1/2}\sigma_h}{|\mathcal{B}_{2,k}|^{1/2}}\Big],
$$

where we use the fact that $\sum_{t=0}^{T-1}(1-\mu) \leq \frac{1}{\mu}$.

Lower bounding the left-hand-side by $\min_{\mathbf{x}}\Phi(\mathbf{x})$ and dividing both sides by $T$, we obtain

$$
\frac{1}{T}\sum_{t=0}^{T-1}\mathbb{E}[\|\nabla\Phi_{1/\bar{\rho}}(\mathbf{w}_t)\|^2]
$$
$$
\leq \frac{2}{\eta T}[\Phi_{1/\bar{\rho}}(\mathbf{x}_0) - \min_{\mathbf{x}}\Phi(\mathbf{x})] + \eta\bar{\rho}M^2 + \frac{\mathcal{C}\bar{\rho}}{T\gamma}\max\{\frac{n}{|\mathcal{B}|}, \frac{\beta m}{|\mathcal{B}_c|}\}
$$
$$
+ \mathcal{C}\Big(\frac{m\beta\eta\bar{\rho}M}{|\mathcal{B}_c|\gamma^{1/2}} + \frac{n^2\eta^2 M^2\bar{\rho}}{|\mathcal{B}|^2\gamma} + \frac{\beta\bar{\rho}\gamma^{1/2}}{|\mathcal{B}_{2,k}|^{1/2}} + \frac{\gamma\bar{\rho}}{|\mathcal{B}_{1,i}|}\Big),
$$

where we use $\gamma_1 = \gamma_2 = \gamma$ and

$$
\mathcal{C} = \max\{4L_{\nabla f}L_g\frac{1}{n}\sum_{i=1}^n\|u_{1,i}^0 - g_i(\mathbf{x}_0)\|^2, 8\frac{1}{m}\sum_{k=1}^m\|u_{2,k}^0 - h_k(\mathbf{x}_0)\|
$$
$$
32L_{\nabla f}L_g^3, 32L_h, 8L_{\nabla f}L_g\sigma_g^2, 16\sigma_h\}.
$$

As we set $M^2 \geq 2L_f^2 L_g^2 + 2\beta^2 L_h^2$ and $2L_{\nabla F} + 2L_{\nabla f}L_f + \beta\frac{\rho_1}{2} + \frac{C}{2} = \frac{\bar{\rho}}{2}$, then with $\gamma = \mathcal{O}(\min\{\frac{|\mathcal{B}_{2,k}|}{\beta^4}\epsilon^4, \frac{|\mathcal{B}_{1,i}|}{\beta}\epsilon^2\})$, $\eta = \mathcal{O}(\min\{\frac{|\mathcal{B}||\mathcal{B}_{1,i}|^{1/2}\epsilon^2}{n\beta^2}, \frac{|\mathcal{B}_c||\mathcal{B}_{2,k}|^{1/2}\epsilon^4}{\beta^5 m}\})$ and $T = \mathcal{O}(\max\{\frac{m\beta^6}{|\mathcal{B}_{2,k}|^{1/2}|\mathcal{B}_c|\epsilon^6}, \frac{n\beta^3}{|\mathcal{B}||\mathcal{B}_{1,i}|^{1/2}\epsilon^4}, \frac{n\beta^2}{|\mathcal{B}||\mathcal{B}_{1,i}|\epsilon^4}\})$, we have

$$
\frac{1}{T}\sum_{t=0}^{T-1}\mathbb{E}[\|\nabla\Phi_{1/\bar{\rho}}(\mathbf{x}_t)\|^2] \leq \mathcal{O}(\epsilon^2).
$$

By Jensen's inequality, we can get $\mathbb{E}[\|\nabla\Phi_{1/\bar{\rho}}(\mathbf{x}_{\hat{t}})\|] \leq \mathcal{O}(\epsilon)$, where $\hat{t}$ is selected uniformly at random from $\{1,\ldots,T\}$.

$\square$

