# OpenReview forum: "Single-loop Algorithms for Stochastic Non-Convex Optimization with Weakly-Convex Constraints"
_TMLR — Accepted by TMLR_

### Review · Reviewer_2i7i · 2025-11-03

**Summary Of Contributions:**

This paper considers an important constrained optimization problem where both the objective and constraint functions are weakly convex.

Strengths:
The proposed approach employs a hinge-based penalty, which permits the use of a constant penalty parameter, enabling us to achieve a state-of-the-art complexity for finding an approximate KKT solution.

 Weaknesses:
1. The authors may have overstated the novelty and achievement of their theoretical complexity bounds. The paper claims to achieve an SOTA complexity for finding an approximate KKT solution. However, the complexity of $\mathcal{O}(1/\epsilon^6)$ achieved by the proposed algorithm has also been established in prior works for similar problem settings. For example, A single-loop SPIDER-type stochastic subgradient method for expectation-constrained nonconvex nonsmooth optimization. CoRR abs/2501.19214 (2025) .

2. The result presented in Theorem 4.2 (or the main convergence theorem) should rigorously state that the output $\mathbf{x}$ is a nearly $\epsilon$-KKT solution in expectation, since the objective function may be stochastic.

**Audience:**

No

**Audience Explanation:**

This paper is very similar to the paper,  A single-loop SPIDER-type stochastic subgradient method for expectation-constrained nonconvex nonsmooth optimization. CoRR abs/2501.19214 (2025) .

**Claims And Evidence:**

Yes

**Claims Explanation:**

The conclusions of Theorem 4.2 and Theorem 5.3 support the claims made in the paper.

**Requested Changes:**

The submission requires several critical revisions concerning its theoretical claims and rigor. First, the comparison in the related works section is inaccurate: the authors must correct their description of Liu and Xu (2025), which also considers multiple constraint functions via a vector-valued function. Relatedly, the paper must include Liu and Xu (2025) in the complexity comparison (e.g., in Table 1), as its $\mathcal{O}(1/\epsilon^6)$ rate matches the claimed state-of-the-art result. Furthermore, for theoretical correctness, all definitions and theorem conclusions (especially Theorem 4.2 and Theorem 5.3) concerning the approximate KKT solution must be qualified by "in expectation," given the stochastic setting. Finally, the authors must resolve the inconsistency between the conclusions of Theorem 4.3 and Theorem 5.3 (one requiring an $\epsilon$-stationary solution and the other achieving only a nearly $\epsilon$-stationary solution), which currently renders the proof chain for the final $\epsilon$-KKT solution non-rigorous.

A minor suggestion is to motivate the choice of the MSVR technique over SPIDER and to formalize the notation in Assumption 5.2.

---

> ### Author Response · Authors · 2025-11-23
>
> **Q1:** The authors may have overstated the novelty and achievement of their theoretical complexity bounds.The complexity of  achieved by the proposed algorithm has also been established in prior works for similar problem settings. For example, A single-loop SPIDER-type stochastic subgradient method for expectation-constrained nonconvex nonsmooth optimization. CoRR abs/2501.19214 (2025) .
>
> **A:** We respectfully disagree with the reviewer. Our work is concurrent to Liu & Xu (2025). Although it appears on arxiv earlier than ours, but our work has been submitted to ICML 2025 for review at the same time.
>
> In addition,  we have explicitly acknowledged their work in the main text (see the last paragraph of Section 2), where we cite Liu & Xu (2025) and discuss the key distinctions between our method and theirs.
>
> Specifically, while both works achieve comparable complexity guarantees, our contribution lies in a different algorithmic design and problem setting: We directly operate on the hinge penalty instead of a smoothed surrogate.
>
> Most importantly, (i) we do not need periodically large batches required in their method employing SPIDER/SARAH-type estimators; (ii) we only need to sample $O(1)$ number of constraint functions for processing at each iteration, while their method require processing all constraint functions at each iteartion.  These design choices offer practical efficiency gains, especially in large-scale stochastic settings.
>
> **Q2:**  First, the comparison in the related works section is inaccurate: the authors must correct their description of Liu and Xu (2025), which also considers multiple constraint functions via a vector-valued function. Relatedly, the paper must include Liu and Xu (2025) in the complexity comparison (e.g.$\mathcal{O}(1/\epsilon^6)$, in Table 1), as its rate matches the claimed state-of-the-art result.
>
>
>
> **A:** Even though they consider multiple constraint functions via a vector-valued function, but their algorithm does not support stochastic update one a sampled constraint function. This is important in machine learning approaches in our experiments. For example, for learning with ROC fairness constraints, we have more than 15 constraints. On the large-scale CheXpert dataset, we only sample 15 constraints per-iteration.  Using all constraints is not efficient as noted in last 4 lines in page 8 for the baseline method ICPPAC.
>
> We will add the comparison in Table 1 in the revision.
>
> **Q3:** The result presented in Theorem 4.2 (or the main convergence theorem) should rigorously state that the output  is a nearly-KKT solution in expectation, since the objective function may be stochastic. Finally, the authors must resolve the inconsistency between the conclusions of Theorem 4.2 and Theorem 5.3 (one requiring an $\epsilon$-stationary solution and the other achieving only a nearly $\epsilon$-stationary solution), which currently renders the proof chain for the final $\epsilon$-KKT solution non-rigorous.
>
>
> **A:** We thank the reviewer for the thoughtful comment. We would like to clarify that  Theorem 4.2 is a general result that does not rely on the stochastic  nature of the algorithm. In particular, Theorem 4.2 shows that any  $\epsilon$-stationary solution of the Moreau-enveloped objective $\Phi_\theta(x)$  is a nearly $\epsilon$-KKT solution of the original constrained problem.  This implication holds regardless of whether the $\epsilon$-stationary point is  obtained deterministically or in expectation.
>
> In the stochastic setting considered in Section 5, Theorem 5.3 proves that our  algorithm produces an iterate $x_{\hat t}$ satisfying $\mathbb{E}\|\nabla \Phi_\theta(x_{\hat t})\| \le \epsilon$.  Combining Theorems 4.2 and 5.3 (as noted in the remark following Theorem 5.3), we therefore conclude that the algorithm output is a nearly  $\epsilon$-KKT solution in expectation for the original constrained problem.  The detailed argument is provided at the end of Appendix D.
>
> To make this more  explicitly, we will add a corollary  following Theorem 5.3 in the revised version by  stating that $x_{\hat t}$ is a nearly $\epsilon$-KKT solution where condition (i) holds in expectation, while conditions (ii) and (iii) hold with high probability for the original problem (1). And if $|h_k(\bar{x}_{\hat t})| < +\infty$, the conditions (ii) and (iii) hold in expectation as showed in the end of Appendix D.
>
> Lastly, there is no inconsistency between Theorem 4.2 and Theorem 5.3. Theorem 4.2 requires an $\epsilon$-stationary solution of the Moreau envelope $\Phi_\theta(\cdot)$, and Theorem 5.3 finds a solutoin $x_{\hat t}$ that is an $\epsilon$-stationary solution of the Moreau envelope $\Phi_\theta(\cdot)$ in expectation.

---

> > ### Author Response · Authors · 2025-11-23
> >
> > **Q4:** A minor suggestion is to motivate the choice of the MSVR technique over SPIDER and to formalize the notation in Assumption 5.2.
> >
> > **A:** We thank the reviewer for the helpful suggestion.  We agree and will make the notation in Assumption 5.2 more formal in the new version. Regarding the use of MSVR: (1) MSVR estimator is specifically designed for estimating many sequences while only accecssing $O(1)$ sequence for update. This makes it greatly suitable for us to handle many constraint functions. In contrast, the SPIDER estimator requires accessing all sequences at the same iteration, which is not efficient for handling many constraint functions.   (2) As discussed in our paper and related works, the SPIDER estimator periodically draws a large batch of samples for estimating the constraint function (see Eq. (3.1) in Liu & Xu, 2025), which increases the per-iteration cost. In contrast, MSVR does not require any large batch sizes.  This design substantially reduces the computational overhead and accelerates the training process, as also demonstrated in Jiang et al., 2022.

---

> > > ### Author Response · Authors · 2025-12-03
> > >
> > > We thank Reviewer 2i7i for the helpful and constructive comments. Following your suggestions, we have made several revisions in the new version (all changes highlighted in blue and marked with “Reviewer 2i7i”)
> > >
> > > **1. Comparison with Liu & Xu (2025) and motivate the choice of the MSVR technique over SPIDER.**
> > > We added remarks after Corollary 5.4 and enhanced the discussion in the Related Work section to clearly compare our method with Liu & Xu (2025), including noting that both achieve the same $\mathcal{O}(1/\epsilon^6)$ complexity but differ in algorithmic design and constraint-sampling strategy. We added the comparison in Table 1 in the revision.
> > >
> > > **2. The stochastic nearly–$\epsilon$-KKT guarantee.**
> > >
> > > We added a new corollary after Theorem 5.3 explicitly stating that the algorithm output satisfies the nearly–$\epsilon$-KKT conditions (condition (i) holds in expectation, while conditions (ii) and (iii) hold with high probability for the original problem (1)).
> > >
> > > **3. Formalize the notation in Assumption 5.2.**
> > >
> > > We made the notation in Assumption 5.2 more formal in the new version, thanks.

---

> > ### Comment · Reviewer_2i7i · 2025-12-04
> > **Thanks for your reply.**
> >
> > Thanks for the rebuttals. My concerns have been addressed. I hope this paper can be accepted.

---

### Review · Reviewer_Mi55 · 2025-11-05

**Summary Of Contributions:**

The paper proposes a single-loop stochastic algorithm for solving non-convex constrained optimization problems where both the objective and constraint functions are weakly convex and possibly non-smooth. The authors base their approach on an exact penalty formulation using a hinge penalty rather than the more common squared-hinge penalty. They argue that this non-smooth penalty preserves weak convexity, allows a constant penalty parameter instead of a large one growing with $1/\epsilon$, leads to a theoretical complexity of $\mathcal{O}(\epsilon^{-6})$ for obtaining a $\epsilon$-KKT point, which is a competitive good rate matching the best double-loop algorithms, and extends naturally to finite-sum coupled compositional objectives. The complexity improvement over existing literatures mainly by using a hinge penalty that keeps $\beta$ constant, avoiding the blow-up in weak convexity parameters; and using variance-reduced stochastic gradients in a single-loop structure. Two applications are reported: Fair learning with ROC-based fairness constraints, and continual learning with non-forgetting constraints. Empirical results compare the hinge-based penalty against squared-hinge and double-loop baselines.

**Audience:**

Yes

**Audience Explanation:**

The topic of stochastic constrained optimization with weakly convex objectives is timely and relevant to the optimization and machine learning theory community, which forms part of TMLR’s readership. The paper builds on a well-studied framework single-loop algorithms. While the theoretical development represents an incremental step beyond existing results and the assumptions could be further relaxed for broader applicability, the work nonetheless contributes to refining our understanding of this line of methods. The experimental validation, though limited in scope, provides some empirical insight into the algorithm’s behavior. Overall, the paper will likely be of interest to readers already following advances in stochastic nonconvex optimization.

**Broader Impact Concerns:**

Not applicable.

**Claims And Evidence:**

Yes

**Claims Explanation:**

Most of the results with proofs are correct and well supported by evidence. I have the following comments.

1. The central theorem (Theorem 4.2) relies on a strong regularity condition (Eq. 3), which effectively assumes a uniform lower bound on the gradients of the constraints across all infeasible points. I believe that relaxing Eq. 3 to impose only an upper bound $c$ is a more reasonable assumption. As the authors mention, under this relaxation the algorithm can be initialized within the feasible region and proceed with carefully chosen step sizes so that all intermediate iterates remain within $V_c$, as adopted in (Huang & Lin, 2023). Could the authors clarify how this relaxation affects the final convergence rate, and whether it is one of the reasons their method achieves better complexity than (Huang & Lin, 2023)?

2. Although Algorithm 1's penalty gradient $G_2^t$ is, in principle, an unbiased estimator of the sampled constraints' gradient (as it is computed only over the active mini-batch $B_c^t$), the proposed estimation strategy may introduce considerable instability. The main concern is that the non-smooth hinge derivative $[u_{2,k}^t]'$ is evaluated using a noisy stochastic estimate $u_{2,k}^t$ rather than the true constraint value $h_k(x_t)$. Near the feasibility boundary, where $h_k(x_t) \approx 0$, small fluctuations in $u_{2,k}^t$ can cause oscillations around zero, making $[u_{2,k}^t]'$ switch between 0 and 1. This behavior can result in oscillatory and inconsistent constraint enforcement, thereby degrading both convergence stability and feasibility control. The issue may be amplified when the number of constraints $m$ is large, since most stored estimators $u_{2,k}$ become stale at any given iteration. Could you comment on this?

3. The algorithm’s step size $\eta$, inner rate $\gamma_2$, and iteration number $T$ all depend explicitly on the unknown target accuracy $\epsilon$. This makes the complexity guarantee non-constructive and the method difficult to implement in practice, as the required parameters cannot be chosen without knowing $\epsilon$ in advance.

**Requested Changes:**

1. The paper needs to explicitly specify the regularity assumptions for each constraint function $h_k$, clarifying when differentiability is required and when only subgradients are available. In several parts of the analysis, the notation $\nabla h_k$ and $\nabla H(x)$ is used informally, even though the stated assumptions (e.g., weak convexity and potential non-smoothness) suggest that subdifferentials would be more appropriate. The authors may replace occurrences of $\nabla h_k$ and $\nabla H(x)$ with the corresponding subdifferential forms $\partial h_k(x)$ and $\partial H(x)$, respectively, and to clearly state the conditions under which these subdifferentials coincide with classical gradients.

2. There is a disconnect between the deterministic notion of a nearly $\epsilon$-KKT solution defined in Definition 3.3 and the stochastic guarantees actually established in Appendix D. The definition requires deterministic satisfaction of all KKT-like conditions, whereas the algorithmic analysis only proves them in expectation or with high probability.

3. The fairness and continual-learning experiments are presented only through training curves without runtime comparisons. Important implementation details (batch sizes, $\gamma$ parameters, and $\beta$ selection) are omitted or inconsistent between text and pseudocode.

---

> ### Author Response · Authors · 2025-11-23
>
> **Q1:** The central theorem (Theorem 4.2) relies on a strong regularity condition (Eq. 3), which effectively assumes a uniform lower bound on the gradients of the constraints across all infeasible points. I believe that relaxing Eq. 3 to impose only an upper bound  is a more reasonable assumption.
>
>   **A:** Thank you for the insightful comment. We would like to clarify that assuming only an upper bound on the constraint gradients is insufficient for establishing convergence. In particular, when the initial point lies in an infeasible region where the gradients of the constraint functions vanish, the algorithm cannot make any progress and will get stuck at such a stationary infeasible point. This issue reflects a fundamental challenge in non-convex constrained optimization: without a condition such as Eq. (3), the problem becomes ill-posed and is generally unsolvable using first-order methods. We note that this limitation is not specific to our method but is shared by all penalty-based approaches, including recent works such as Alacaoglu & Wright (2024), Li et al. (2024a), and Li et al. (2024b).
>
>  **Q2:** As the authors mention, under this relaxation the algorithm can be initialized within the feasible region and proceed with carefully chosen step sizes so that all intermediate iterates remain within $V_{c}$, as adopted in (Huang & Lin, 2023). Could the authors clarify how this relaxation affects the final convergence rate？
>
>   **A:** Thank you for the question. This relaxation does not affect the final convergence rate.
>
> In our analysis, we assume that (3) holds over the entire set $\mathcal{V}$, our algorithm can start from an infeasible point, and the non-vanishing constraint gradients guarantee that the iterates make progress toward feasibility.
>
> If we relax the assumption to $V_c := \{x : c \ge \max_k h_k(x) > 0\}$, then the assumption becomes **strictly weaker**. Under this relaxation, the algorithm can be initialized within the feasible region and proceed with step sizes chosen carefully to ensure that all intermediate iterates remain within $\mathcal{V}_c$. Note that even if some iterates become slightly infeasible, as long as the step size is sufficiently small, the gradients of the constraint functions will guide the iterates back toward feasibility. This way,  Equation (3) continues to hold within $\mathcal{V}_c$, we can still conduct exactly the same convergence analysis just select feasible initial point $\mathbf{x}_0$ and proceed with step sizes chosen carefully.
>
>
>  **Q3:** Whether it is one of the reasons their method achieves better complexity than (Huang & Lin, 2023)?
>
>
>  **A:**  No, this is not one of the reasons for the improved complexity. The purpose of adopting the relaxation used in Huang & Lin (2023) is  an alternative way to weaken Assumption (3). Our improved complexity is due to using a different method and algorithm. Huang & Lin (2023) use a classical switching subgradient–type method within a proximal-point framework, whereas our approach directly solves the hinge-penalty reformulation. In particular, the exact penalty function $\frac{\beta}{m}\sum_{k=1}^m[h_k(\mathbf{x})]_+$ is a special case of a non-smooth weakly-convex finite-sum compositional objective of the form considered in  Hu et al. (2024) as we view hinge function as an outer function. We also use the variance reduction technique MSVR for estimating constraint functions as illustrated in Section 5, which helps us achieve a better convergence rate.

---

> > ### Author Response · Authors · 2025-11-23
> >
> > **Q4:** Although Algorithm 1's penalty gradient is $G_2^t$, in principle, an unbiased estimator of the sampled constraints' gradient (as it is computed only over the active mini-batch $B_c^t$), the proposed estimation strategy may introduce considerable instability. The main concern is that the non-smooth hinge derivative
> > is evaluated using a noisy stochastic estimate rather than the true constraint value $h_k(x_t)$. Near the feasibility boundary, where $h_k(x_t)\approx 0$, small fluctuations in $u_{2,k}^t$ can cause oscillations around zero, making $[u_{2,k}^t]_+^{'}$
> > switch between 0 and 1.
> >
> > This behavior can result in oscillatory and inconsistent constraint enforcement, thereby degrading both convergence stability and feasibility control. The issue may be amplified when the number of constraints $m$ is large, since most stored estimators $u_{2,k}$  become stale at any given iteration. Could you comment on this?
> >
> >
> >  **A:** **First**, our analysis (Eqn. [13]) shows that as long as $\sqrt{\gamma_2} \to 0$, $\eta / \sqrt{\gamma_2} \to 0$, and $T \to \infty$,  the error $|u_{2,k}^t - h_k(x_t)|$ converges to zero. This guarantees that the solution converges to a KKT point asymptotically.
> >
> > **Second**, for the non-asymptotic case with $\epsilon > 0$, we agree that $u_{2,k}^t$ may fluctuate around zero. However, such oscillations can be effectively controlled by using sufficiently small $\eta$ and $\gamma_2$, ensuring they do not interfere with achieving the condition $h_k(x_t) \le \epsilon$. In our experiments, we did not observe any instability in constraint enforcement. Please refer to Fig. 3 (bottom) and Fig. 4.
> >
> >
> >  **Q5:** The algorithm’s step size $\eta$, inner rate $\gamma_2$, and iteration number $T$ all depend explicitly on the unknown target accuracy $\epsilon$ . This makes the complexity guarantee non-constructive and the method difficult to implement in practice, as the required parameters cannot be chosen without knowing  in advance.
> >
> >
> >
> > **A** Thank you for the comment. While we state the convergence results for a given $\epsilon$, it is also possible to state the convergence for a given $T$. In this case, we can let $\eta, \gamma_2$ depend on $T$, which can yiled the same complexity result. In practice, these hyper-parameters are alawys tuned to achieve the best performance. For example, in the first experiment we choose the initial learning rate from $\{10^{-3}, 10^{-4}\}$ and apply a standard decay schedule; in the second experiment we use the same magnitude choices as in Li et al.(2024a). These settings are fully reported in the experimental section, and our results show that they work well in practice.
> >
> >
> > **Q6:**  The paper needs to explicitly specify the regularity assumptions for each constraint function $h_k$, clarifying when differentiability is required and when only subgradients are available. The authors may replace occurrences of $\nabla h_k$ and $\nabla H(x)$ with the corresponding subdifferential forms $\partial h_k$ and $\partial H(x)$, respectively, and to clearly state the conditions under which these subdifferentials coincide with classical gradients.
> >
> >
> > **A:** We are sorry for the confusion! We state at the begining of Section 4.1 *For simplicity of exposition and comparison  with prior works, we consider differentiable constraint functions. The results also hold for the non-smooth case, with the gradient replaced by the subgradient.*
> >
> > Indeed, Lemma 4.3 can be modified as $h(x) - \min_y h(y)\leq \frac{1}{2\mu}\text{dist}(0, \partial h(x))^2$. The proof is the same. In fact, the example below Lemma 4.3 is for a non-smooth non-differntiable function (at $x=1,-1$). Lemma 4.4 can be modified as $\text{dist}(0, \lambda_{\min}(\partial H(x)))\geq \sigma$, where $\partial H$ denotes the set of sub-Jacobian matrices and $\lambda_{\min}(\partial H(x))$ denotes the set of their minimum singular values. The proof can be easily modified to prove the condition (3). We will modify the statements in the revision.
> >
> >
> >
> >
> >  **Q7:** There is a disconnect between the deterministic notion of a nearly $\epsilon$-KKT solution defined in Definition 3.3 and the stochastic guarantees actually established in Appendix D. The definition requires deterministic satisfaction of all KKT-like conditions, whereas the algorithmic analysis only proves them in expectation or with high probability.
> >
> >
> > **A:** We respectfully disagree with the reviewer. For any stochastic algorithm, convergence guarantees are necessarily stated either in expectation or with high probability. However, the definition of the target solution itself (e.g., an $\epsilon$-stationary point in standard stochastic optimization) is always deterministic.

---

> > > ### Author Response · Authors · 2025-11-23
> > >
> > > **Q8:**  The fairness and continual-learning experiments are presented only through training curves without runtime comparisons. Important implementation details (batch sizes,  parameters, and  selection) are omitted or inconsistent between text and pseudocode.
> > >
> > >
> > >  **A:** Thank you for the comment. We want to clarify that our goal in the fairness and continual-learning experiments is to compare the performances of different methods in reducing the  objective value when the constraints are satisfied. Moreover, for completeness, we have reported the approximately runtimes in Appendix B of the supplementary material. As stated there, each AUC-maximization experiment with ROC fairness constraints take approximately 4 hours per seed, and the continual-learning experiments with non-forgetting constraints require about 12 hours per seed. These runtimes are comparable across all methods since they share identical model architectures and computational setups.
> > >
> > >  We would also like to clarify that both the fairness and continual-learning experiments follow a shared parameter configuration across all methods, including the learning-rate schedule, weight decay, batch sizes, and the selections of $|\mathcal{B}_c|$ and $|\mathcal{B}_k|$. We tune the penalty coefficient $\beta$, which allows us to study how different methods reduce constraint violations and how their performances in the objective value change when the constraints are satisfied. This consistent setup provides a fair and controlled comparison of the optimization behavior of all methods.
> > >
> > >
> > > We will make this reference clearer in the revised version.

---

> > > > ### Author Response · Authors · 2025-12-03
> > > >
> > > > We thank Reviewer Mi55 for the helpful and constructive comments.
> > > > Following your suggestions, we have made several revisions in the new version (all changes highlighted in blue and marked with “Reviewer Mi55”).
> > > >
> > > > **1. The necessity of Assumption (3).**
> > > >
> > > > We added a more discussion in the remark following Theorem 4.2 explaining why Assumption (3) is standard and necessary for ensuring progress from infeasible points.
> > > >
> > > > **2.Clarify the effect of relaxing Eq. (3).**
> > > >
> > > >  We updated the remark Eq. (3) to explain that relaxing the assumption to
> > > > $V_c :=$ {$x : c \ge \max_k h_k(x) > 0$} does not change the final convergence rate.
> > > >
> > > > **3.Discuss the stability of the hinge-derivative estimator.**
> > > >
> > > > We added a discussion in Section 5 describing the behavior of $u_{2,k}^t$
> > > >   near the feasibility boundary and clarifying the estimator remains stable in practice.
> > > >
> > > > **4.Clarify dependence of step sizes on $\epsilon$**
> > > >
> > > > We added a brief remark following Theorem 5.3 explaining how the same complexity holds when convergence is stated for a fixed number of iterations and our experiments show that these settings work well in practice.
> > > >
> > > >  **5. Fix differentiability vs. subgradients.**
> > > >
> > > > We revised Lemma 4.3, Lemma 4.4, and the corresponding notation in the proofs.
> > > >
> > > >
> > > > **6. Runtime and implementation details.**
> > > > We stated the runtime information and added the discussion of the shared parameter configurations in Appendix A.

---

> > > > > ### Comment · Reviewer_Mi55 · 2025-12-05
> > > > > **Thanks**
> > > > >
> > > > > Thank you to the authors for thoroughly addressing my questions. I have reviewed the updated manuscript and find that most of my concerns have been resolved. I am therefore in favor of accepting the paper.

---

### Review · Reviewer_oUVS · 2025-11-09

**Summary Of Contributions:**

This paper proposes a single-loop penalty-based stochastic algorithm for optimization problem where both the objective and the inequality constraints are weakly-convex and potentially non-smooth. The idea is to use a non-smooth, hinge-based penalty function, which permits the use of a constant penalty parameter,  and achieve a complexity of $\mathcal{O}(\epsilon^{-6})$ for finding a near $\epsilon$-KKT solution.

**Audience:**

Yes

**Audience Explanation:**

For optimization researchers,  the main contribution is a single-loop stochastic algorithm that achieves an $\mathcal{O}(\epsilon^{-6})$ complexity for non-smooth, weakly-convex constrained problems. The novelty lies in using a hinge penalty and only requires a constant parameter rather than one that must go to infinity, which is an interesting result.

The analysis under non-smooth settings is also useful for applied researchers. It directly addresses the structure of many modern ML problems.

**Broader Impact Concerns:**

No concerns

**Claims And Evidence:**

Yes

**Claims Explanation:**

The complexity analysis is standand. I checked all the proofs in the appendix, and do not see any major issues.

**Requested Changes:**

1. The estimation of the stochastic gradient is not clear now, please make it clearer. For example, the $\mathcal{B}_c^t$ should be a set, but in the Algorithm1, there is "$\mathcal{B}_c^t \in${1,...,m} ".

2. The end of the proof of Thm 5.3 seems to use Eq. (3) as an assumption when proving condition (ii)

3.  Why "since the objective and the constraint functions are non-smooth, finding an ϵ-KKT solution is not
tractable", could you involve some insights and discussion? And if the objective and constraint functions are smooth, is there some way to convert a near-stationary point to a stationary point？

4. Considering that the main contribution of this work is the use of a penalty method with a constant penalty parameter, there are some works that use primal-dual type methods, which also permit a constant penalty parameter. I believe it is worth discussing these works, for example:

[1]Wenqiang Pu, Kaizhao Sun, and Jiawei Zhang. Smoothed proximal lagrangian method for nonlinear constrained programs, 2024	arXiv:2408.15047

[2]Ruichuan Huang, Jiawei Zhang, and Ahmet Alacaoglu. Stochastic smoothed primal-dual algorithms for nonconvex optimization with linear inequality constraints. In Forty-second International Conference on Machine Learning, 2025.

Minor thing:

5. In the comparison with the work by Liu & Xu (2025), there is 'they consider only one constraint function'. However, I checked that in Liu & Xu (2025), they use a vector-valued function.

---

> ### Author Response · Authors · 2025-11-23
>
> **Q1:** The estimation of the stochastic gradient is not clear now, please make it clearer. For example, the $\mathcal{B}_t$ should be a set, but in the Algorithm1, there is $\mathcal{B}_t\in$ {1,...,m} ".
>
>  **A:** Thank you for your careful reading and helpful suggestions. We agree and will improve the writing and the clarity in the revision.
>
>
>  **Q2:** The end of the proof of Thm 5.3 seems to use Eq. (3) as an assumption when proving condition (ii).
>
>  **A:** Yes, at the end, the proof of Theorem 5.3 indeed assumes Equation (3).  Specifically,  we first suppose $\max_k h_k( \bar{\mathbf{{x}}}_{\hat{t}})> \epsilon$.
>
> Then we drive (24) based on (3). Then (24) leads to a contradiction with the selection of $\beta$. Hence, we can prove that it holds with probability $1-\mathcal{O}(\epsilon)$ that $\max_k h_k( \bar{\mathbf{{x}}}_{\hat{t}})\leq \epsilon$, meaning that condition(ii) holds with probability $1-\mathcal{O}(\epsilon)$. Then with the selection of $\lambda_k$, condition (iii) also holds with probability $1-\mathcal{O}(\epsilon)$.
>
>
>   **Q3:** Why "since the objective and the constraint functions are non-smooth, finding an ϵ-KKT solution is not tractable", could you involve some insights and discussion?
>
>
> **A:**  Thank you for asking us to clarify this point. When $f(\cdot)$ is non-smooth, finding an $\epsilon$-stationary solution such that  $\|\nabla f(w)\|\le \epsilon$ is difficult even for a convex, unconstrained function.
> Let us consider a simple example $\min_w |w|$ as we point out in main text.  The only stationary point is the optimal solution $w_* = 0$, and any $w \neq 0$ is not an $\epsilon$-stationary solution ($\epsilon < 1$) no matter how close $w$ to $0$. In other words, unless the iterate generated by the algorithm can exactly land on $w=0$, it will not produce an $\epsilon$-stationary point.
>
>
> To address this issue, an effective approach for solving non-smooth optimization is to approximate the original problem by a smoothed one using different smoothing techniques, including Nesterov’s smoothing (Yu Nesterov. S, 2005), randomized smoothing (Guy Kornowski and Ohad Shamir. 2022), and Moreau envelope (Damek Davis, 2019). Based on  Moreau envelopes, Ma et al. (2020); Boob et al. (2023); Huang & Lin (2023); Jia & Grimmer (2022) have developed algorithms and theories for problems with non-smooth objective and constraint functions, which derives a nearly $\epsilon$-KKT solution.
>
> We will add more discussion of this point.
>
> Yu Nesterov, 2025. Smooth minimization of non-smooth functions. Math. Program., 103(1):127–152, May 2005.
>
> Guy Kornowski and Ohad Shamir, 2022. Oracle complexity in nonsmooth nonconvex optimization. Journal of Machine Learning Research, 23(314):1–44, 2022.
>
>
>
>
>  **Q4:** And if the objective and constraint functions are smooth, is there some way to convert a near-stationary point to a stationary point？
>
> **A:** Thank you for your insightful question. The answer is yes. Suppose the objective and constraint functions are smooth,  using the Lipschitz contiuity of the gradients, and that of constraint functions, it is easy to prove that  a nearly $\epsilon$-stationary point is also an $O(\epsilon)$-stationary point. This is because $\|x-\bar x\|_2\leq \epsilon$ and leveraging the Lipschitz conunity condition, every condition in the definition of nearly $\epsilon$-KKT solution can be converted to that on $x$. We will add more  discussion of this point in the new version.

---

> > ### Author Response · Authors · 2025-11-23
> >
> > **Q5:** Considering that the main contribution of this work is the use of a penalty method with a constant penalty parameter, there are some works that use primal-dual type methods, which also permit a constant penalty parameter. I believe it is worth discussing these works, for example:
> >
> > [1]Wenqiang Pu, Kaizhao Sun, and Jiawei Zhang. Smoothed proximal lagrangian method for nonlinear constrained programs, 2024 arXiv:2408.15047
> >
> > [2]Ruichuan Huang, Jiawei Zhang, and Ahmet Alacaoglu. Stochastic smoothed primal-dual algorithms for nonconvex optimization with linear inequality constraints. In Forty-second International Conference on Machine Learning, 2025.
> >
> > **A:** Thank you for pointing out these two relevant works and for highlighting recent progress on smoothed proximal and primal–dual methods with constant penalty parameters. Both Pu et al. (2024) and Huang et al. (2025) make valuable contributions in convex or linearly constrained settings.
> >
> > Our work addresses a different problem setting for stochastic non-convex objectives together with **non-convex constraints**. Our algorithm does not involve any dual variables or primal–dual updates. In this sense, our approach is structurally distinct from the smoothed primal–dual frameworks considered in these papers. We will add more discussions about the difference between our method and primal-dual method.
> >
> > It is indeed an interesting future direction to explore whether primal–dual techniques can be extended to fully non-convex constrained optimization in the stochastic setting, potentially achieving even better convergence guarantees. We will mention this perspective in the revised version.
> >
> >
> >  **Q6:** In the comparison with the work by Liu & Xu (2025), there is 'they consider only one constraint function'. However, I checked that in Liu & Xu (2025), they use a vector-valued function.
> >
> >
> > **A:** The difference is that Liu & Xu treat multiple constraint as a single one, i.e., they need to compute mini-batch estimators of  all constraint functions at each iteration. In contrast, we do sampling on the constraint functions, and only require estimating $O(1)$ constraint functions at each iteration, which is achieved by using MSVR estimator.

---

> > > ### Author Response · Authors · 2025-12-03
> > >
> > > We thank Reviewer oUVS for the helpful and constructive comments. Following your suggestions, we have made several revisions in the new version (all changes highlighted in blue and marked with “Reviewer oUVS”)
> > >
> > >  **1. Clarify the estimation of the stochastic gradient.**
> > >
> > >  We corrected the notation in Algorithm 1 and Algorithm 2 so that $\mathcal{B}_t^c$ is explicitly presented as a set.
> > >
> > > **2. Add a discussion on non-smoothness and ε-KKT tractability.**
> > >
> > > On page 4, we added a simple example explaining why finding an ε-KKT solution is not tractable when both the objective and constraint functions are non-smooth.
> > >
> > > **3. Address the smooth-case question.**
> > > At the end of page 4, we added a Remark explaining that, in the smooth setting with Lipschitz continuous gradients, a nearly $O(\epsilon)$-stationary point can be converted into an $O(\epsilon)$-stationary point.
> > >
> > > **4. Add some discussion on primal–dual methods with constant penalty parameters.**
> > >
> > > In Section 7 *Conclusion and Discussion*, we added a brief discussion comparing our hinge-based exact penalty approach with primal–dual methods that also use constant penalty parameters (e.g., Pu et al., 2024; Huang et al., 2025), and  point this is an interesting future direction to explore the primal-dual techniques can be extended to fully non-convex constrained optimization in the stochastic setting.
> > >
> > > **5. Clarify the comparison with Liu & Xu (2025).**
> > >
> > > In the Related Work section, we update a clearer explanation of the difference between our method and Liu & Xu (2025).

---

> > > > ### Comment · Reviewer_oUVS · 2025-12-07
> > > > **Thanks for your rebuttals**
> > > >
> > > > Thanks for your rebuttals. The authors have effectively addressed my previous comments. I will support the acceptance of this paper.

---

### Comment · Reviewer_2i7i · 2025-12-15
**Thanks for your rebuttals**

Thanks for the rebuttals. My concerns have been addressed. I hope this paper can be accepted.

---

### Decision · Action_Editor_c6Sf · 2026-01-08

**Recommendation:** Accept as is

**Additional Comments:**

In your edits right now, you mention reviewer aliases from openreview when you respond to their comments. Please remove reviewer aliases from your pdf.

**Audience:**

Yes

**Audience Explanation:**

This paper studies stochastic constrained optimization, which is widely encountered in machine learning, including important applications such as continual learning and fairness-constrained optimization. As a result, this work will interest the audience.

**Claims And Evidence:**

Yes

**Claims Explanation:**

This paper focuses on stochastic constrained optimization with weakly convex objective functions and weakly convex inequality constraints. The authors present a penalty-based single-loop method that achieves the best-known $O(1/{\epsilon^6})$ sample complexity in this setting. A key technical contribution is the use of a hinge-loss–based penalty for inequality constraints, which enables the use of a constant penalty parameter and avoids the slow convergence typically caused by large penalty values. Compared to existing works in the literature, the proposed method simultaneously avoids the complex double-loop structures commonly used in prior approaches, requires only an $O(1)$ batch size instead of large batches, and samples only $O(1)$ constraints at each iteration rather than all constraints. The assumptions rely on commonly used regularity conditions. The theoretical analysis is correct, and the experimental results are convincing. All reviewers agree that this work fills an interesting and previously underexplored gap in the literature, and that its claims are supported by accurate, clear, and convincing evidence. Overall, the positioning of the work and the presentation of the ideas are very well executed.